# Parallel processing, hierarchical transformations, and sensorimotor associations along the 'where' pathway

**Raymond Doudlah[1†], Ting-Yu Chang[2†], Lowell W Thompson[1], Byounghoon Kim[1], Adhira Sunkara[3], Ari Rosenberg[1]\***

[1]Department of Neuroscience, University of Wisconsin-Madison, Madison, United States; [2]National Defense Medical Center, Taipei, Taiwan; [3]WiSys Technology Foundation, Madison, United States

**Abstract** Visually guided behaviors require the brain to transform ambiguous retinal images into object-level spatial representations and implement sensorimotor transformations. These processes are supported by the dorsal 'where' pathway. However, the specific functional contributions of areas along this pathway remain elusive due in part to methodological differences across studies. We previously showed that macaque caudal intraparietal (CIP) area neurons possess robust 3D visual representations, carry choice- and saccade-related activity, and exhibit experience-dependent senso-rimotor associations (Chang et al., 2020b). Here, we used a common experimental design to reveal parallel processing, hierarchical transformations, and the formation of sensorimotor associations along the 'where' pathway by extending the investigation to V3A, a major feedforward input to CIP. Higher-level 3D representations and choice-related activity were more prevalent in CIP than V3A. Both areas contained saccade-related activity that predicted the direction/timing of eye movements. Intriguingly, the time course of saccade-related activity in CIP aligned with the temporally integrated V3A output. Sensorimotor associations between 3D orientation and saccade direction preferences were stronger in CIP than V3A, and moderated by choice signals in both areas. Together, the results explicate parallel representations, hierarchical transformations, and functional associations of visual and saccade-related signals at a key juncture in the 'where' pathway.

**\*For correspondence:**
ari.rosenberg@wisc.edu

[†]These authors contributed equally to this work

## Editor's evaluation

This study compares the neuronal activity of two interconnected cortical areas in the dorsal visual pathway, V3A and CIP, during perceptual decisions based on judging the tilt of 3D visual patterns. This is a novel comparison between neural activity in these two cortical areas during perceptual decisions and gives insight into the hierarchical relationship and roles of these two areas. CIP shows higher-order spatial representations and more choice-correlated responses. Furthermore, the study finds modulation of V3A activity by extraretinal factors, suggesting that V3A be better characterized as contributing to higher-order behavioral functions as opposed to low-level visual feature processing.

## Introduction

The 3D perceptual and sensorimotor capabilities of primates facilitate their ability to shape the world. For instance, 3D spatial reasoning is a key predictor of engineering problem-solving ability (***Hsi et al., 1997***). These capabilities are supported by the dorsal 'where' pathway. In particular, high-level visual transformations are thought to occur in brain areas located at the parieto-occipital junction (***Tsao***

*et al., 2003*; *Chang et al., 2020b*). Parietal cortex is thought to then implement sensorimotor transformations that map those sensory representations to motor responses (*Pause and Freund, 1989*; *Rushworth et al., 1997*; *Buneo and Andersen, 2006*). However, assigning particular functions to specific areas has been challenging due to methodological differences across studies. Here, we used a common experimental design to investigate two areas that bridge the parieto-occipital junction in macaque monkeys: intermediate visual area V3A and the caudal intraparietal (CIP) area.

Area CIP is a site of 3D visual processing (*Taira et al., 2000*; *Tsutsui et al., 2002*; *Rosenberg et al., 2013*; *Rosenberg and Angelaki, 2014a*; *Rosenberg and Angelaki, 2014b*), which is functionally correlated (*Tsutsui et al., 2003*; *Elmore et al., 2019*) and causally linked (*Tsutsui et al., 2001*; *Van Dromme et al., 2016*) to 3D perception. Saccade-related activity and sensorimotor associations in CIP may further support goal-directed behaviors (*Chang et al., 2020b*) via connections to oculomotor and prehensile areas (*Lewis and Van Essen, 2000*; *Premereur et al., 2015*; *Van Dromme et al., 2016*; *Lanzilotto et al., 2019*).

By comparison, V3A findings have been highly conflicting. Some imply relatively low-level image processing such as spatiotemporal filtering (*Gaska et al., 1987*; *Gaska et al., 1988*), basic stereoscopic depth selectivity (*Anzai et al., 2011*), and 2D direction selectivity (*Nakhla et al., 2021*). Other findings link V3A to high-level processes underlying stable, allocentric representations of the world. This includes combining visual and extraretinal signals to represent objects in non-retinal coordinates (*Galletti and Battaglini, 1989*; *Galletti et al., 1990*; *Sauvan and Peterhans, 1999*; *Nakamura and Colby, 2002*), distinguishing veridical object motion from self-induced retinal image motion (*Galletti et al., 1990*), and 3D spatial processing (*Tsao et al., 2003*; *Elmore et al., 2019*). Furthermore, V3A activity is modulated by attention and memory-related factors, and some neurons show postsaccadic activity (*Nakamura and Colby, 2000*).

To directly compare the functional properties of these interconnected areas, we used a common experimental design to assess (i) selectivity for the 3D pose (orientation and position) of planar surfaces, (ii) choice-related activity during a 3D orientation discrimination task (*Chang et al., 2020a*), (iii) saccade-related activity during a visually guided saccade task (*Munoz and Wurtz, 1995*; *Hanes and Schall, 1996*), and (iv) sensorimotor associations (*Chang et al., 2020b*). Multiple lines of evidence converged to support a V3A-to-CIP hierarchy. First, our findings revealed that robust 3D pose representations were most prominent in CIP. Second, choice-related activity was associated with robust 3D pose tuning in both areas but most prevalent in CIP. Third, the areas contained similar proportions of neurons with saccade-related activity that predicted the direction and timing of eye movements. Saccade-related activity started earlier in V3A than CIP and the CIP time course closely matched the temporally integrated V3A output, suggesting that saccade-related signals in CIP may originate in V3A. Notably, both areas showed sensorimotor associations (which were stronger in CIP than V3A) that were statistically moderated by choice-related activity. Together, these findings challenge classical notions of sensorimotor dichotomies, argue for a reclassification of V3A as association cortex, and implicate choice-related activity as a novel factor in sensorimotor processing.

## Results

To investigate the contributions of areas V3A and CIP to the transformation of retinal images into object-level representations and goal-directed sensorimotor processing, we compared the 3D selectivity, saccade-related properties, and sensorimotor associations of 692 V3A neurons (Monkey L: N = 311; Monkey F: N = 263; Monkey W: N = 118) and 437 previously analyzed CIP neurons (Monkey L: N = 218; Monkey F: N = 219) (*Chang et al., 2020b*). Areas V3A and CIP were dissociated from each other and adjacent regions using multiple anatomical and functional criteria (*Figure 1*; 'Materials and methods'). Supporting a V3A-to-CIP hierarchy, the median visual response latency was shorter in V3A (46 ms) than CIP (52 ms) and the receptive fields were smaller in V3A than CIP (*Figure 1—figure supplement 1*).

### Behavioral discrimination of 3D surface orientation

To investigate the transformation of visual representations into goal-directed behaviors, we trained three monkeys to report the 3D orientation of a planar surface (*Chang et al., 2020a*). Specifically, they performed an eight-alternative forced choice (8AFC) tilt discrimination task with planar surfaces

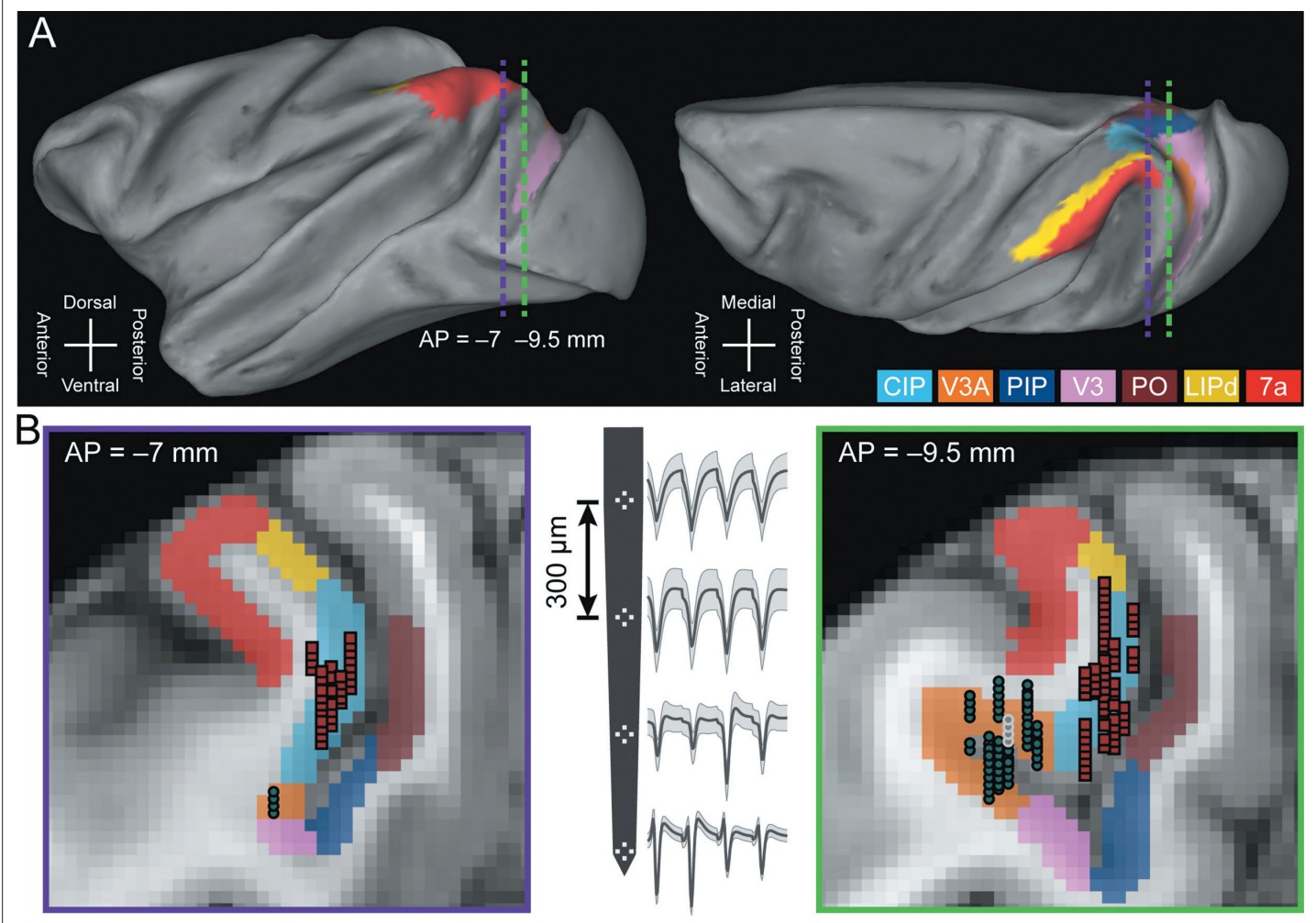

**Figure 1.** Neuronal recordings. (**A**) Lateral (left) and dorsal (right) views of the inflated cortical surface of Monkey L (left hemisphere). Dashed lines mark the coronal sections in (**B**). (**B**) Coronal sections (left: AP = −7 mm; right: AP = −9.5 mm) with MRI-based estimates of the boundaries of V3A, CIP, and adjacent areas. Recording locations for V3A (blue-gray circles) and CIP (red squares) were projected along the AP axis onto the closest of the two coronal sections shown. A schematic of a four-tetrode laminar probe with spike waveforms from the V3A recording marked with white circles in the right coronal section is shown (middle). CIP, caudal intraparietal area (light blue); V3A, visual area V3A (orange); PIP, posterior intraparietal area; V3, visual area V3; PO, parieto-occipital area; LIPd, dorsal aspect of the lateral intraparietal area; 7a, area 7a.

The online version of this article includes the following figure supplement(s) for figure 1:

**Figure supplement 1.** Response latencies and receptive field sizes.

presented at different orientations and distances (*Figure 2*). The orientation was defined by two angular variables (*Stevens, 1983*; *Rosenberg et al., 2013*): tilt and slant. Tilt describes which side of the plane was nearest to the monkey and slant describes the rotation in depth (*Figure 2A*). Planes were presented for 1 s while fixation was maintained on a target at the center of the screen. The monkey then reported the plane's tilt (the near side) via a saccade to the corresponding choice target, regardless of the slant or distance (*Figure 2B*).

Behavioral performance was quantified each session (V3A: Monkey L: N = 39; Monkey F: N = 38; Monkey W: N = 14; CIP: Monkey L: N = 26; Monkey F: N = 27) by calculating the distribution of reported tilt errors (ΔTilt = reported tilt − presented tilt) for each non-zero slant and distance (16 conditions) pooled across tilt (*Chang et al., 2020a*; *Chang et al., 2020b*). Each error distribution was then fit with a von Mises probability density function (*Equation 1*) and behavioral sensitivity was quantified as the concentration parameter ($\kappa$) of the fit. To assess how sensitivity depended on the viewing conditions, we ran a linear mixed-effects model with distance, slant, and area as main effects and animal as a random effect. Consistent with our previous findings, sensitivity decreased with distance

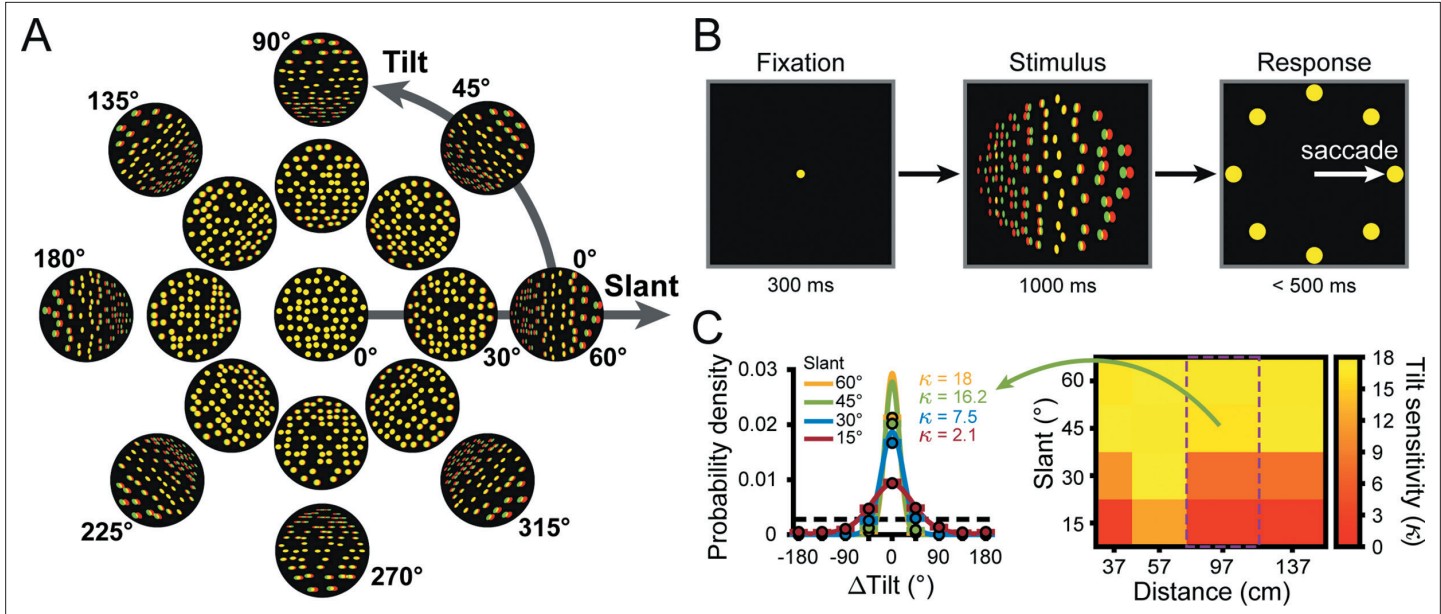

**Figure 2.** Stimuli, task, and behavioral performance. (**A**) Planar surfaces were defined using random dots with perspective and stereoscopic cues (illustrated here as red-green anaglyphs). For clarity, the size and number of dots differ from the actual stimuli. (**B**) Eight-alternative tilt discrimination task. A trial began by fixating a dot at the center of the screen (fixation was always at screen distance, 57 cm) for 300 ms (left). A plane was then presented with a given tilt (0–315°, 45° steps), slant (0–60°, 15° steps), and distance (37, 57, 97, or 137 cm) for 1 s (middle). The fixation target and plane then disappeared and eight choice targets corresponding to the eight tilts appeared (right). This cued the monkey to saccade to one of the targets to report which side of the plane was nearest. (**C**) Behavioral performance. Error distributions of reported tilts for each slant at 97 cm for Monkey W (left). Data points show the mean probability of a given ΔTilt (reported tilt – presented tilt), and error bars show standard error of the mean (SEM) across sessions (N = 14). Solid curves are von Mises probability density functions with sensitivities ($\kappa$) indicated. The black dashed line marks chance level. The heat map (right) shows the mean tilt sensitivity for each slant (rows) and distance (columns) for Monkey W across sessions. Yellow hues indicate higher sensitivities. Green arrow and purple rectangle mark the data shown in the error distribution plots (left).

from fixation and increased with slant (***Figure 2C***). Correspondingly, behavioral sensitivity significantly depended on distance (p=4.1 × 10⁻³⁰) and slant (p=2.2 × 10⁻³⁰⁸). There was no significant effect of area (p=0.46), indicating that behavioral performance was similar during the V3A and CIP sessions. To relate this pattern of behavioral sensitivity to V3A and CIP activity, we next characterized the simultaneously recorded neuronal responses.

## Hierarchical transformations in the representation of 3D orientation

The visual system is thought to turn ambiguous 2D retinal signals into behaviorally relevant 3D object representations through a series of transformations. We therefore hypothesized that CIP would contain a higher-level representation of 3D pose than V3A. For a 3D pose selective neuron, the shape of its 3D orientation tuning curve will be tolerant to distance, but its overall response amplitude (gain) should be distance-dependent (***Janssen et al., 2000***; ***Nguyenkim and DeAngelis, 2003***; ***Alizadeh et al., 2018***; ***Chang et al., 2020b***). In contrast, a neuron selective for lower-level visual features (e.g., binocular disparity) will have 3D orientation tuning curves whose shape and gain are highly distance-dependent. To test for 3D pose tuning, we therefore assessed how 3D orientation tuning depended on distance.

The 3D orientation tuning curves of four representative V3A neurons are shown in ***Figure 3A–D*** (qualitatively similar examples from CIP are shown in Figure 3 of ***Chang et al., 2020b***). Some neurons had similar 3D orientation tuning across all (***Figure 3A***) or most (***Figure 3B***) distances with distance-dependent gain changes, implying 3D pose tuning. Others had significant orientation tuning at a single distance (ANOVA, p<0.05; Bonferroni–Holm corrected for four distances; ***Figure 3C***), which may reflect intermediate selectivity for gradients of absolute binocular disparity (***Nguyenkim and DeAngelis, 2003***). The orientation tuning of other neurons changed substantially with distance (***Figure 3D***), implying lower-level visual feature selectivity. These examples suggest that V3A contains

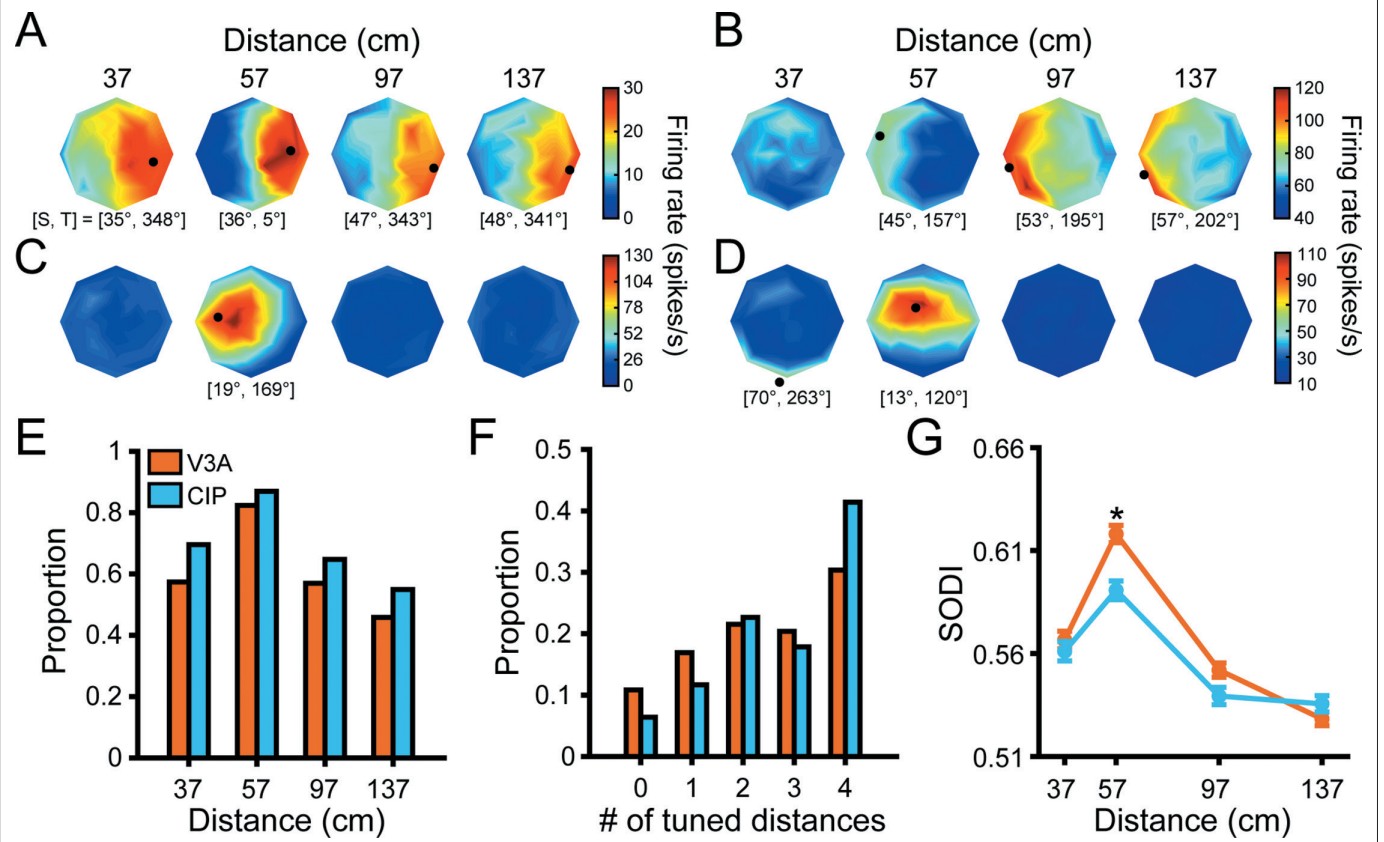

**Figure 3.** Comparison of 3D orientation tuning across distance. (**A–D**) Four example visual area V3A (V3A) neurons from Monkey W (**A, C**) and Monkey F (**B, D**). Heat maps show firing rate plotted as a function of tilt (angular axis) and slant (radial axis). Red hues indicate higher firing rates. Black dots mark preferred 3D orientations from Bingham function fits at distances with significant tuning (ANOVA, p<0.05; Bonferroni–Holm corrected for four distances). Some dots are not located on a disc because the largest tested slant was 60° but slant ranges from 0° to 90°. The preferred slant (S) and tilt (T) are indicated for each tuned distance. (**E**) Proportion of neurons with significant orientation tuning at each distance for V3A (orange; proportions: 37 cm = 0.57, 57 cm = 0.82, 97 cm = 0.57, 137 cm = 0.46) and caudal intraparietal (CIP) area (blue; proportions: 37 cm = 0.70, 57 cm = 0.87, 97 cm = 0.65, 137 cm = 0.55). (**F**) Proportion of neurons with significant orientation tuning at each possible number of distances for V3A (proportions: #0 = 0.11, #1 = 0.17, #2 = 0.22, #3 = 0.20, #4 = 0.30) and CIP (proportions: #0 = 0.06, #1 = 0.12, #2 = 0.23, #3 = 0.18, #4 = 0.41). (**G**) Comparison of surface orientation discrimination index (SODI) values at each distance for V3A (orange) and CIP (blue). Data points and error bars are mean and SEM across neurons with significant orientation tuning, respectively. The asterisk indicates a significant difference between V3A and CIP SODI values at 57 cm only (ANOVA followed by Tukey's HSD test, p<0.05).

The online version of this article includes the following figure supplement(s) for figure 3:

**Figure supplement 1.** Distributions of 3D orientation preferences.

**Figure supplement 2.** Cross-area comparison of 3D orientation tuning curve shape.

a heterogeneous population of neurons whose functional properties range from processing low-level visual features to 3D object pose.

In both areas, more neurons had 3D orientation tuning at 57 cm (fixation distance) than at the other distances (*Figure 3E*). Although the proportion of neurons with significant tuning at each distance was greater in CIP than V3A, the cross-area difference was not significant (chi-squared test, across animals: $\chi^2 = 2.4$, p=0.50; Monkey L: $\chi^2 = 0.62$, p=0.89; Monkey F: $\chi^2 = 1.6$, p=0.67). However, CIP neurons were typically tuned for 3D orientation at more distances than V3A neurons (chi-squared test, across animals: $\chi^2 = 21.2$, p=2.9 × 10$^{-4}$; Monkey L: $\chi^2 = 13.0$, p=0.01; Monkey F: $\chi^2 = 8.7$, p=0.07), implying greater convergence of orientation information across distance within CIP than V3A (*Figure 3F*).

We next examined the 3D orientation preferences and tuning curve shapes by fitting each significant orientation tuning curve (ANOVA, p<0.05; Bonferroni–Holm corrected for four distances) with a Bingham function (*Bingham, 1974*). The Bingham function is a low-dimensional, parametric model over tilt and slant that describes V3A and CIP 3D orientation tuning curves (*Rosenberg et al., 2013*;

*Rosenberg and Angelaki, 2014a*; *Elmore et al., 2019*; *Chang et al., 2020b*). The preferred orientation taken from these fits is marked with a black dot for the example neurons in *Figure 3A–D*. In both V3A and CIP, the full span of 3D orientations was represented at each distance (*Figure 3—figure supplement 1*), indicating that both areas can support neural codes for 3D pose. To compare the shape of the orientation tuning curves, we used the Bingham parameters describing the bandwidth ($\lambda_2$), isotropy ($\lambda_1$), and axis about which any anisotropy occurred ($\Phi$). First, there was a slight but significant tendency for V3A neurons (median $\lambda_2 = 0.80$) to be more narrowly tuned than CIP neurons (median $\lambda_2 = 0.65$; linear mixed-effects model with area, absolute distance from fixation, and animal as fixed effects, and neuron as a random effect, p=$1.5 \times 10^{-3}$; *Figure 3—figure supplement 2A and B*). This difference may reflect convergent input from multiple V3A neurons onto individual CIP neurons. The tuning bandwidths also increased with distance from fixation (p=$6.5 \times 10^{-6}$), implying information loss that mirrored the behavioral finding that tilt discrimination performance decreased with distance from fixation (*Chang et al., 2020a*; *Chang et al., 2020b*; *Figure 2*, see Figure 5). No difference was observed across animals (p=0.77). Second, the V3A tuning curves were less isotropic (more elongated; median $\lambda_1 = -1.62$) than the CIP tuning curves (median $\lambda_1 = -0.92$), and the difference was significant (p=$1.3 \times 10^{-9}$; *Figure 3—figure supplement 2C and D*). The level of anisotropy also significantly increased with distance from fixation (p=$9.4 \times 10^{-6}$). No difference was observed across animals (p=0.78). Lastly, the distributions of $\Phi$ peaked at approximately 90° in both V3A (median $\Phi = 88°$) and CIP (median $\Phi = 89°$), indicating that any anisotropy in the tuning curves generally occurred along the tilt/slant axes (*Figure 3—figure supplement 2E and F*). These findings indicate greater orientation tuning symmetry in CIP than V3A, which may be important for perceptual sensitivity to changes in object orientation to not depend on the axis of rotation (*Chang et al., 2020b*).

A previous study found that 3D orientation was better discriminated based on the responses of individual V3A than CIP neurons (*Elmore et al., 2019*). To follow up on that finding, we computed a surface orientation discrimination index (SODI) that quantifies the difference in responses to preferred and non-preferred orientations relative to the response variability (*Equation 2*). Neurons with stronger 3D orientation selectivity have SODI values closer to one, whereas those with weaker selectivity have values closer to zero. For each neuron, we calculated the SODI at each distance with significant orientation tuning. In both areas, the mean SODI had an inverted U-shape as a function of distance that peaked at 57 cm (fixation distance; *Figure 3G*). This indicates that 3D orientation was most discriminable at the fixation distance, which may be a downstream consequence of V1 neurons tending to prefer smaller binocular disparities (*Prince et al., 2002*). Consistent with the *Elmore et al., 2019* study, which measured 3D orientation tuning at the fixation distance only, we found that the SODI values across animals were significantly larger in V3A than CIP at the fixation distance (ANOVA followed by Tukey's HSD test, p=$1.1 \times 10^{-5}$). However, the differences were not significant at any other distance (p≥0.47). For Monkey L, the SODI values were significantly larger in V3A than CIP at 37, 57, and 97 cm (p≤$8.5 \times 10^{-3}$), but not 137 cm (p=0.95). For Monkey F, they were not significantly different at any distance (p≥0.52). Taken together with the *Elmore et al., 2019* findings, these results are consistent with a cross-area difference that, as we consider next, may reflect a transformation from lower-level visual feature selectivity to higher-level 3D pose tuning.

## Hierarchical refinement of 3D pose representations

To test for cross-area differences in lower-level visual feature selectivity versus higher-level 3D pose tuning, we assessed how the 3D orientation tuning curves depended on distance (*Janssen et al., 2000*; *Nguyenkim and DeAngelis, 2003*; *Alizadeh et al., 2018*). This approach recently revealed 3D pose tuning in CIP (*Chang et al., 2020b*) but has not been applied to V3A. It thus remains unknown if 3D pose tuning in CIP is simply inherited or reflects a qualitative transformation of feedforward input.

To quantify the distance-dependence of 3D orientation tuning curve shape, we fit each 3D pose tuning curve with a separable model (*Equation 3*) and computed a tolerance index (*Chang et al., 2020b*). Tolerance values near zero indicate that the shape of the orientation tuning curve changed substantially with distance (as expected for neurons selective for low-level visual features). Values near one indicate that the shape changed minimally with distance (implying 3D pose tuning). As shown for the example neurons, larger tolerance values were associated with 3D pose tuning (tolerance = 0.96, 0.74; *Figure 3A and B*, respectively), modest values with more intermediate representations (tolerance = 0.41; *Figure 3C*), and low values with low-level feature selectivity (tolerance = 0.17;

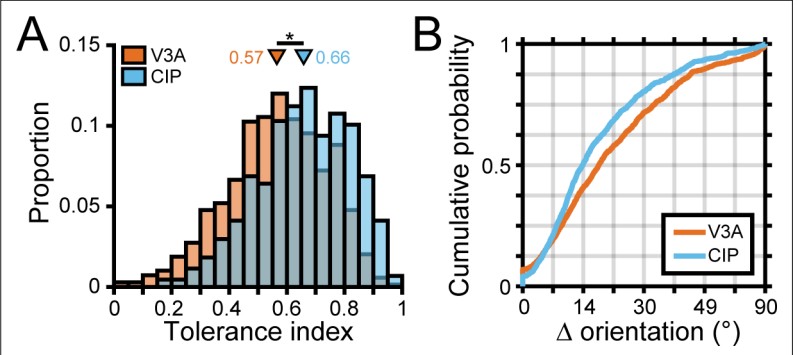

**Figure 4.** Robust 3D pose tuning was less prevalent in visual area V3A (V3A) than the caudal intraparietal (CIP) area. (**A**) Distribution of tolerance values in V3A (orange; N = 692) and CIP (blue; N = 437). Triangles mark mean tolerance values, and the asterisk indicates a statistically significant difference (two-sample *t*-test, p<0.05). (**B**) Cumulative density functions over the angular deviations between the orientation preference at each distance and the principal orientation for each neuron.

*Figure 3D*). Across the V3A population, the tolerance values revealed a heterogeneous population in which neurons ranged from having low-level visual feature selectivity to high-level 3D pose tuning (*Figure 4A*, orange bars).

To test our hypothesis that 3D pose tuning would be more prevalent in CIP than V3A, we performed two complementary analyses. First, we evaluated if there was a cross-area difference in the extent to which the shape of the 3D orientation tuning curves depended on distance by comparing the tolerance distributions. In V3A, the mean tolerance was $0.57 \pm 6.7 \times 10^{-3}$ SEM across animals (*Figure 4A*, orange bars; N = 692). Individually, the mean tolerances in V3A were $0.56 \pm 8.5 \times 10^{-3}$ (Monkey L, N = 311), $0.58 \pm 0.01$ (Monkey F, N = 263), and $0.55 \pm 0.02$ (Monkey W, N = 118). In CIP, the mean tolerance was $0.66 \pm 7.7 \times 10^{-3}$ SEM across animals (*Figure 4A*, blue bars; N = 437). Individually, the mean tolerances in CIP were $0.65 \pm 0.01$ (Monkey L, N = 218) and $0.66 \pm 0.01$ (Monkey F, N = 219). As predicted, the tolerance values in CIP were significantly larger than in V3A (two-sample *t*-test, across animals: $p=7.4 \times 10^{-19}$; Monkey L: $p=4.7 \times 10^{-11}$; Monkey F: $p=3.7 \times 10^{-7}$), indicating that the shape of 3D orientation tuning curves was more similar across distance in CIP than V3A. Second, we evaluated if there was a cross-area difference in the extent to which the orientation preferences (independent of other tuning parameters such as bandwidth) depended on distance. For each neuron, we calculated the angular deviation between the preferred orientation at each distance and the principal orientation (*Chang et al., 2020b*; 'Materials and methods'). Across neurons, we then computed cumulative density functions over the angular deviations (*Figure 4B*) and found that the deviations were

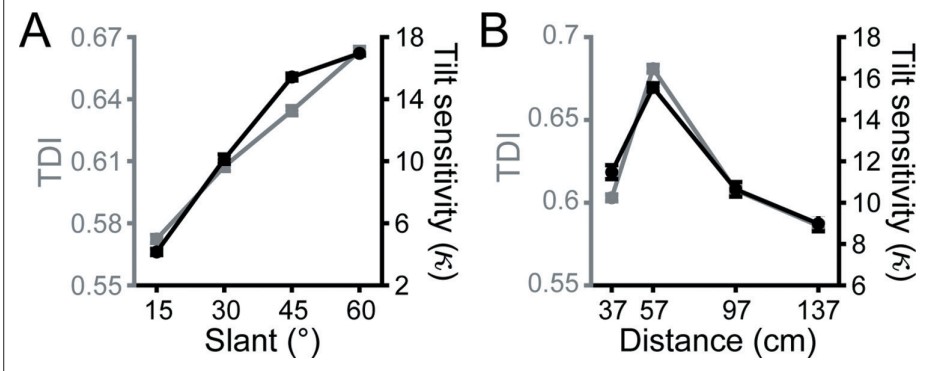

**Figure 5.** Neuronal correlates of tilt sensitivity. (**A**) Mean visual area V3A (V3A) tilt discrimination index (TDI) (gray) and behavioral tilt sensitivity (black) increased with slant. TDI values (and behavioral sensitivities) were averaged across neurons (or monkeys) and distances. (**B**) Mean V3A TDI and behavioral tilt sensitivity had an inverted U-shape relationship with distance. TDI values (behavioral sensitivities) were averaged across neurons (monkeys) and slants. Error bars are SEM.

significantly greater in V3A than CIP (Kolmogorov–Smirnov test, across animals: p=1.2 × 10⁻⁹; Monkey L: p=5.1 × 10⁻⁶; Monkey F: p=5.9 × 10⁻⁴). Thus, although both areas represented the full span of 3D orientations at each distance (*Figure 3—figure supplement 1*), the orientation preferences of individual neurons were more similar across distance in CIP than V3A. These analyses together suggest that a transformation from lower-level visual features to higher-level 3D object representations occurs between V3A and CIP.

## Neuronal correlates of behavioral tilt sensitivity

We previously found that behavioral tilt sensitivity, which increases as a function of slant and has an inverted U-shape pattern as a function of distance from fixation (*Chang et al., 2020a*; *Figure 2C* and black curves in *Figure 5*), is correlated with neuronal tilt discriminability in CIP (*Chang et al., 2020b*). To test if a functional correlation between behavior and neuronal activity also exists for V3A, we calculated a tilt discrimination index (TDI; *Equation 2*) at each slant–distance combination for each neuron, following *Chang et al., 2020b*. Analogous to the SODI, the TDI quantifies the difference in responses to preferred and non-preferred tilts relative to the response variability. Notably, the mean TDI values in V3A followed the same trends as the behavioral sensitivity for slant (*Figure 5A*) and distance (*Figure 5B*). Indeed, the behavioral tilt sensitivities and TDI values were highly correlated across all 16 slant–distance combinations (Monkey L: Spearman $r = 0.92$, p=2.2 × 10⁻³⁰⁸; Monkey F: $r = 0.98$, p=2.2 × 10⁻³⁰⁸; Monkey W: $r = 0.74$, p=1.5 × 10⁻³). Thus, neuronal tilt discriminability in both V3A and CIP was functionally correlated with the 3D tilt sensitivities of the monkeys across a wide range of viewing conditions.

## V3A carries choice-related activity during 3D orientation discrimination

Previous studies found that roughly half of CIP neurons carried choice-related activity during 3D orientation discrimination tasks (*Elmore et al., 2019*; *Chang et al., 2020b*). In contrast, one of those studies also reported that choice-related activity was essentially non-existent in V3A, but only tested 23 neurons (*Elmore et al., 2019*). Given that choice-related activity is preferentially carried by CIP neurons with robust 3D pose tuning (*Chang et al., 2020b*), the dearth of V3A choice-related activity in the Elmore study could have occurred if the small sample mostly included neurons with low-level feature selectivity. Because that study was not designed to distinguish between low-level visual feature selectivity and 3D pose tuning, we wanted to reassess if V3A carries choice-related activity.

To dissociate choice-related and orientation-selective activity, we analyzed responses to frontoparallel planes (S = 0° and tilt undefined, making them task ambiguous) only. To remove any distance-related response differences, we z-scored the responses at each distance and then pooled across distance. The responses were then grouped according to the monkey's reported tilt. We first computed eight population-level time courses aligned to the tilt choice that elicited the maximum response for each neuron (*Figure 6A*). Following an initial transient response, the eight time courses began to separate and showed parametric tuning with amplitudes that symmetrically fell off from the preferred choice (note the similarity of the ±45°, ±90°, and ±135° time courses), thus revealing choice-related activity in V3A. The onset of choice-related activity was defined as the first time point that the eight time courses significantly diverged (191 ms; ANOVA, p<0.05). In contrast, the onset of choice-related activity in CIP was 202 ms (see Figure 5A in *Chang et al., 2020b*). The finding that choice-related activity appeared first in V3A may reflect that choice signals in CIP contain a large bottom-up contribution from V3A.

Across the populations, 172 (25%) V3A and 201 (46%) CIP neurons carried significant choice-related activity (ANOVA, p<0.05; *Figure 6B*, inset bar plot). The proportion of neurons with choice-related activity varied across monkeys but was always more prevalent in CIP (Monkey L: N = 72, 33%; Monkey F: N = 129, 59%) than V3A (Monkey L: N = 23, 7%; Monkey F: N = 127, 48%; Monkey W: N = 22, 19%). To test if the strength of choice-related activity differed between the areas, we computed a choice discrimination index (CDI; *Equation 2*) for each neuron with significant choice tuning (Appendix 1 shows that this index is unaffected by z-scoring, which was used in calculating the choice tuning curves). The CDI values were highly similar in V3A (mean CDI = 0.38 ± 5.6 × 10⁻³ SEM, N = 172) and CIP (mean CDI = 0.38 ± 5.7 × 10⁻³ SEM, N = 201) and not significantly different (Wilcoxon rank-sum test, p=0.20). Thus, although there was a cross-area difference in the prevalence of choice-related activity, it was nevertheless present in V3A and similar in strength to CIP.

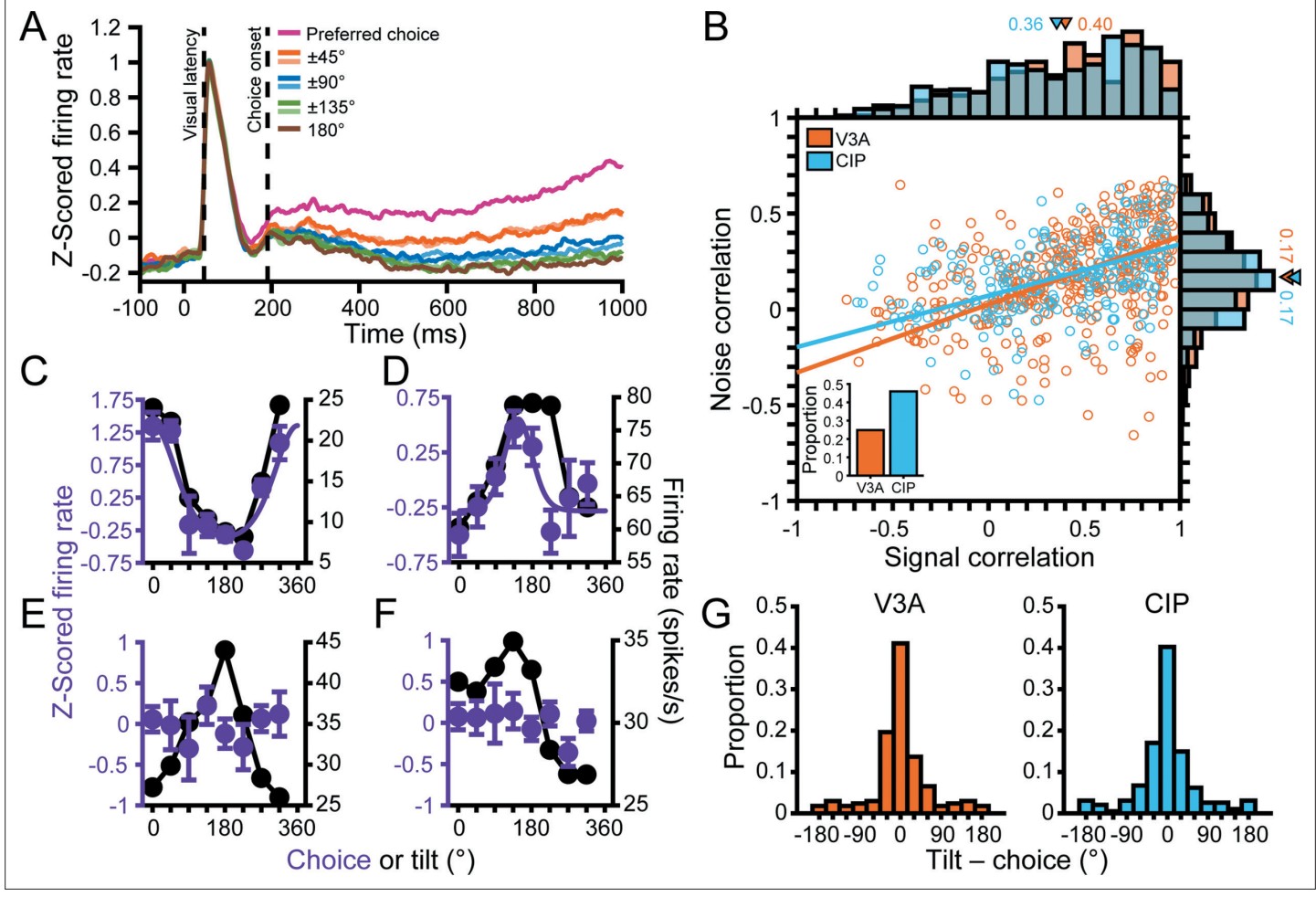

**Figure 6.** Choice tuning in visual area V3A (V3A) and its relationship to tilt tuning. (**A**) Population time courses for each choice option relative to the preferred choice. Curves show z-scored responses averaged over neurons. Stimulus onset = 0 ms. Vertical dashed lines mark the median visual response latency (46 ms) and the onset of choice-related activity (191 ms). (**B**) Comparison of noise and signal correlations in V3A (orange; N = 404 pairs) and the caudal intraparietal (CIP) area (blue; N = 244 pairs). Solid lines show type II linear regression fits. Marginal histograms show the distributions of noise (right) and signal (top) correlations. Triangles mark mean values. Inset shows proportion of neurons with choice-related activity in V3A (25%) and CIP (46%). (**C–F**) Choice tuning curves (left axis, purple) and tilt tuning curves marginalized over slant and distance (right axis, black) for the four example V3A neurons from **Figure 3**. Data points show mean firing rate, and error bars are SEM. Solid purple curves are von Mises fits for neurons with significant choice tuning (ANOVA, p<0.05). Black lines are linear interpolations. (**G**) Comparison of preferred surface tilt and choice preferences in V3A (left, N = 168) and CIP (right, N = 194). The peaks near 0° indicate that the preferences generally aligned. Bars at ±180° are identical.

One potential reason that choice-related activity was more prevalent in CIP than V3A is a difference in the structure of correlated variability between the areas (***Nienborg and Cumming, 2006***). To test this, we computed noise and signal correlations using the 3D pose data for pairs of neurons simultaneously recorded on the same tetrode ('Materials and methods'). The noise correlations were almost identical in V3A (across animals: mean r = 0.17 ± 0.01 SEM, N = 404 pairs; Monkey L: 0.15 ± 0.01, N = 206; Monkey F: 0.15 ± 0.03, N = 121; Monkey W: 0.25 ± 0.03, N = 77) and CIP (across animals: 0.17 ± 0.01, N = 244 pairs; Monkey L: 0.14 ± 0.02, N = 107; Monkey F: 0.19 ± 0.02, N = 137), and not significantly different (Wilcoxon rank-sum test, across animals: p=0.62; Monkey L: p=0.58; Monkey F: p=0.49; **Figure 6B**, right marginal histogram). However, noise correlations alone are not sufficient to account for differences in choice-related activity. Instead, the relationship between noise and signal correlations matters (***Liu et al., 2013***; ***Cumming and Nienborg, 2016***). We therefore quantified the signal correlations for the same pairs of neurons and found that they were also similar in V3A (across animals: mean r = 0.40 ± 0.02 SEM; Monkey L: 0.32 ± 0.03; Monkey F: 0.41 ± 0.04; Monkey W: 0.61 ± 0.04) and CIP (across animals: 0.36 ± 0.03; Monkey L: 0.26 ± 0.04; Monkey F:

0.43 ± 0.03), and not significantly different (across animals: p=0.15; Monkey L: p=0.32; Monkey F: p=0.86; *Figure 6B*, top marginal histogram). We then compared the relationship between noise and signal correlations (*Figure 6B*). As expected, higher noise correlations were associated with higher signal correlations (*Shadlen et al., 1996*; *Cohen and Maunsell, 2009*; *Gu et al., 2011*). Importantly, the linear relationship between noise and signal correlation was not significantly different between the areas (ANCOVA, across animals: p=0.37; Monkey L: p=0.06; Monkey F: p=0.30). Furthermore, between pairs of neurons within each area in which (i) both had choice-related activity, (ii) neither had choice-related activity, or (iii) one had but the other did not have choice-related activity, we found no significant differences in the magnitudes of noise (Kruskal–Wallis test, both p≥0.15) and signal (both p≥0.19) correlations or their linear relationship (ANCOVA, both p≥0.23). These results suggest that differences in the structure of correlated variability either within or across areas did not account for the presence of choice-related activity. As such, the cross-area difference in prevalence of choice-related activity was consistent with a stronger functional correlation between neuronal activity and 3D perceptual decisions in CIP than V3A.

Choice tuning curves are shown for the four example V3A neurons from *Figure 3* in *Figure 6C–F*. The two neurons with higher tolerances (*Figure 3A and B*) both carried choice-related activity (*Figure 6C and D*, purple curves). Notably, the tilt tuning curves (marginalized over slant and distance; *Figure 6C and D*, black curves) were well aligned with the choice tuning curves. In contrast, the neurons with intermediate and low tolerances (*Figure 3C and D*) did not carry choice-related activity (*Figure 6E and F*, purple points). To quantify the relationship between tilt and choice preferences for neurons with significant orientation and choice tuning, we took the circular difference between each neuron's preferred tilt (from the principal orientation) and preferred tilt choice (from the von Mises fit). In both areas, the median circular difference between the preferences (V3A: 1.57°, N = 168; CIP: –0.75°, N = 194) was not significantly different from 0° (circular median test, both p≥0.49), indicating that the tilt and choice preferences tended to align (*Figure 6G*). To assess if the relationship between tilt and choice preferences differed between the areas, we compared the widths of the two distributions and found that they had similar circular variance (CV) values (V3A: CV = 0.34; CIP: CV = 0.33), which were not significantly different (two-sample concentration difference test, p=0.88) (*Fisher, 1995*). Thus, the strength of the association between tilt and choice preferences was similar in V3A and CIP.

## Choice-related activity was associated with robust 3D tuning

We previously found that CIP neurons with more robust 3D pose tuning (higher tolerance values) preferentially carried choice-related activity (*Chang et al., 2020b*). To assess this relationship in V3A, we compared the tolerance values of V3A neurons with and without choice-related activity (*Figure 7A*). Indeed, V3A neurons with choice-related activity had a mean tolerance of 0.65 ± 1.3 × 10$^{-2}$ SEM (N =

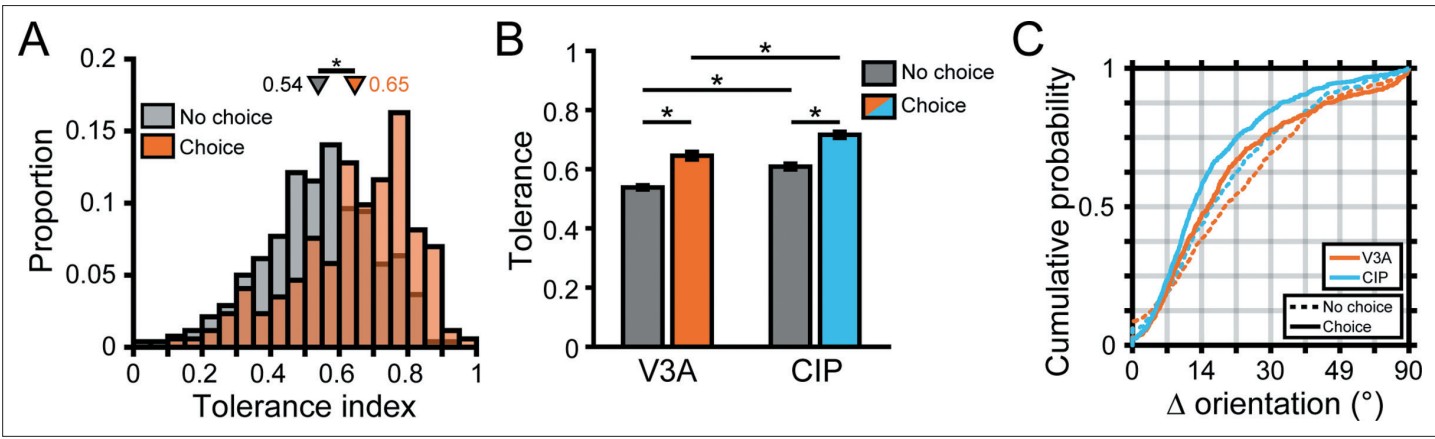

**Figure 7.** Robust 3D pose tuning was associated with choice-related activity. (**A**) Distribution of tolerance values for visual area V3A (V3A) neurons with (orange bars, N = 172) and without (gray bars, N = 520) choice-related activity. Triangles mark mean tolerances. (**B**) Cross-area comparison of tolerance values for neurons with (colored bars) and without (gray bars) choice-related activity. The V3A data in (**A**) is reproduced in (**B**) for comparison with the caudal intraparietal (CIP) area. Bar height indicates mean tolerance, and error bars are SEM. Horizontal lines and asterisks indicate significant differences in (**A, B**) (ANOVA followed by Tukey's HSD test, p<0.05). (**C**) Cumulative density functions over the angular deviations between the orientation preference at each distance and the principal orientation for neurons with (solid lines) and without (dashed lines) choice-related activity.

172), whereas those without choice-related activity had a mean tolerance of $0.54 \pm 7.2 \times 10^{-3}$ SEM (N = 520), and the difference was significant (ANOVA followed by Tukey's HSD test, $p=3.8 \times 10^{-9}$). Thus, choice-related activity was preferentially carried by V3A neurons with more robust 3D pose tuning, as in CIP.

Because more robust 3D pose tuning was associated with choice-related activity, we were concerned that the difference in 3D selectivity between V3A and CIP (*Figure 4*) simply reflected a cross-area difference in the prevalence of choice signals. However, this was not the case. First, tolerance values were greater in CIP than V3A both for neurons with (ANOVA followed by Tukey's HSD test, $p=1.7 \times 10^{-7}$) and without ($p=1.6 \times 10^{-4}$) choice-related activity (*Figure 7B*). Second, comparisons of the cumulative density functions over the angular deviations between orientation preferences at each distance and the principal orientation (*Figure 7C*) revealed significantly smaller deviations for neurons with than without choice-related activity in both areas (Kolmogorov–Smirnov test, $p \leq 1.1 \times 10^{-5}$). The deviations were also significantly smaller in CIP than V3A both for neurons with ($p=1.3 \times 10^{-4}$) and without ($p=8.8 \times 10^{-4}$) choice-related activity. Thus, 3D pose tuning was more robust in CIP than V3A, regardless of whether the neurons carried choice-related activity.

## V3A carries saccade-related activity

Previous studies reported that V3A contains extraretinal signals associated with creating stable, allocentric representations of the world (*Galletti and Battaglini, 1989*; *Galletti et al., 1990*; *Sauvan and Peterhans, 1999*; *Nakamura and Colby, 2002*), as well as postsaccadic activity (*Nakamura and*

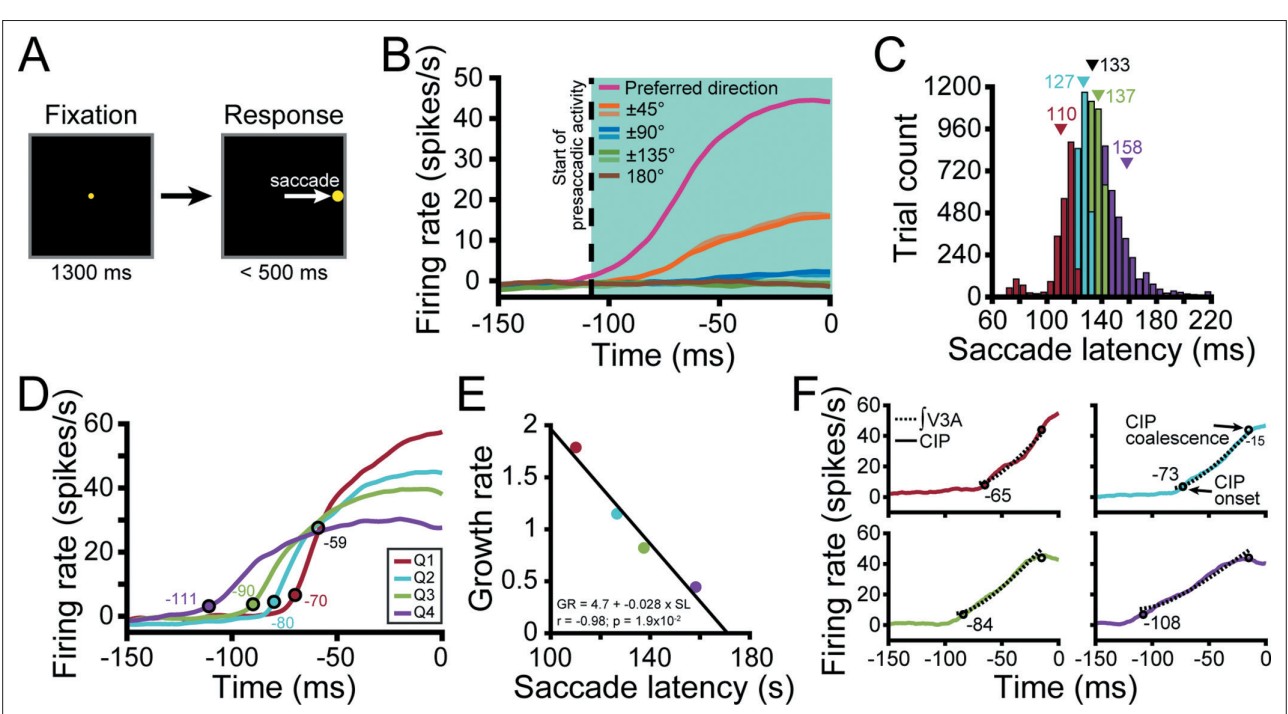

**Figure 8.** Saccade-related activity in visual area V3A (V3A). (**A**) Visually guided (pop-up) saccade task. A target was fixated for 1.3 s (matching the tilt discrimination task duration; left) after which it disappeared and a single saccade target appeared at one of eight locations (matching the choice targets in the tilt discrimination task; right). A saccade was then made to the target. (**B**) Population time courses for each saccade direction relative to the preferred direction. Curves show responses averaged over neurons. Saccade initiation = 0 ms. Vertical dashed line marks the start of saccade-related activity (–108 ms). (**C**) Histogram of saccade latencies divided into quartiles (Q). Triangles mark mean values (black for the full distribution). (**D**) Time courses of saccade-related activity conditioned on the saccade latency quartile. Colored circles mark the start of V3A activity for each quartile (ANOVA, p<0.05). Open black circle marks the point at which the V3A curves approximately coalesced. (**E**) Inverse linear relationship between the growth rate (GR) of saccade-related activity and mean saccade latency (SL) for each quartile. (**F**) The temporally integrated V3A time courses for each quartile (dashed curves) were well-aligned to the observed caudal intraparietal (CIP) area time courses (solid curves). Circles mark the start of CIP activity for each quartile and the point at which the curves approximately coalesced.

The online version of this article includes the following figure supplement(s) for figure 8:

**Figure supplement 1.** Comparison of saccade-related activity and responses to visual flashes without eye movements.

*Colby, 2000*). Recently, we discovered saccade-related activity in CIP that predicted the direction and timing of eye movements (*Chang et al., 2020b*). We therefore hypothesized that V3A may possess similar saccade-related activity. To test this possibility, we trained the monkeys to perform a visually guided (pop-up) saccade task (*Munoz and Wurtz, 1995*; *Hanes and Schall, 1996*; *Figure 8A*).

To first determine if V3A carries saccade-related activity, we computed eight population-level time courses relative to the saccade direction that elicited the maximum response for each neuron (*Figure 8*). Noting that the time courses were parametrically tuned, we calculated the start of the activity by finding the first time point at which they significantly diverged (ANOVA, p<0.05). Intriguingly, saccade-related activity started in V3A (a classically defined 'intermediate' visual area) 108 ms prior to saccade initiation, which was earlier than in CIP (102 ms; see Figure 8B in *Chang et al., 2020b*). In both areas, the time course of saccade-related activity was distinct from the visually evoked response measured during receptive field mapping, consistent with presaccadic activity (*Figure 8— figure supplement 1*; see 'Discussion').

The finding of saccade-related activity in V3A prompted us to test if that activity predicted the timing of the saccades. Each trial in which a saccade was made in the preferred saccade direction was labeled with the saccade latency (the time from target appearance to saccade initiation). The distribution of latencies was then divided into quartiles (*Figure 8C*) and the time course of saccade-related activity computed for each quartile (*Figure 8D*). On trials that the saccade latency was shorter, saccade-related activity started closer to the saccade initiation. Indeed, the growth rate (linear slope) from the start of saccade-related activity (ANOVA, p<0.05) to when the four curves approximately coalesced (59 ms before saccade initiation; 'Materials and methods') was highly correlated and inversely related to the mean saccade latency ($r = -0.98$, $p=1.9 \times 10^{-2}$; *Figure 8E*).

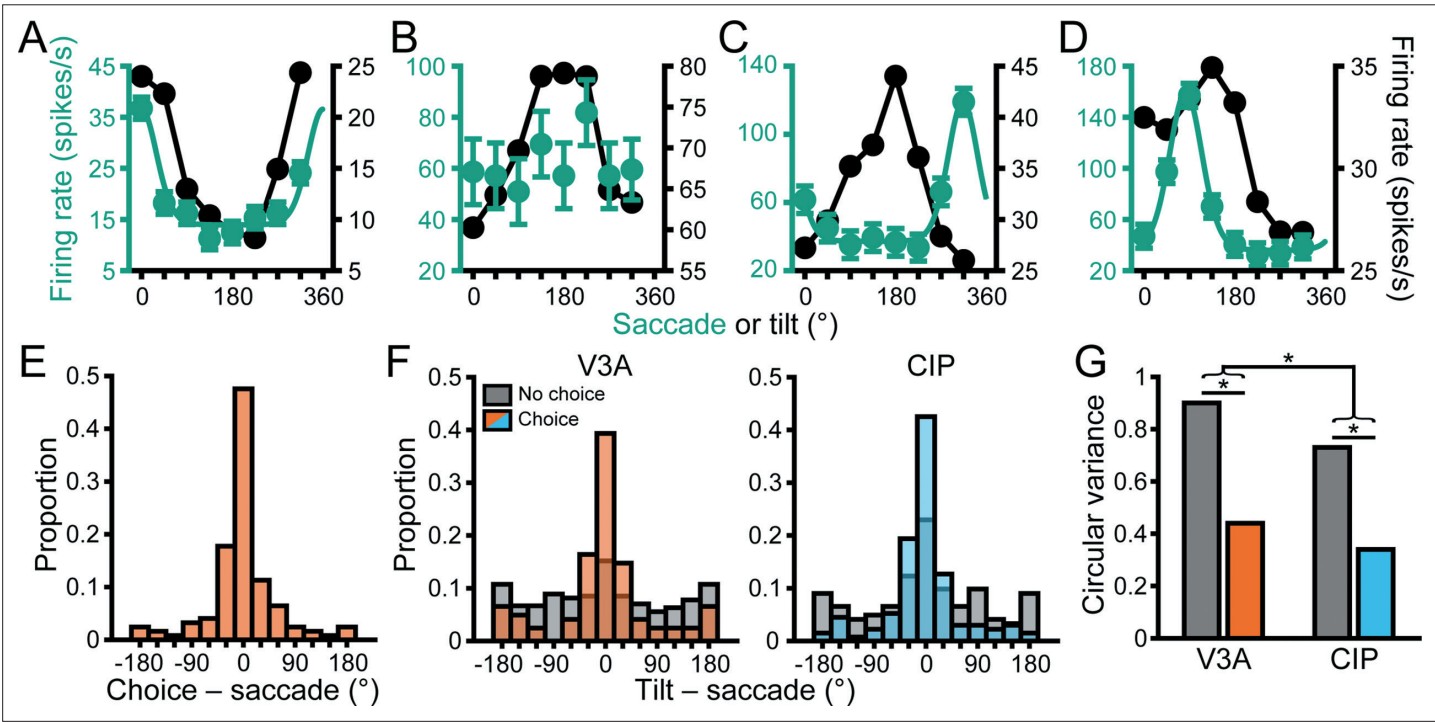

**Figure 9.** Sensorimotor associations were moderated by choice-related activity. (**A–D**) Saccade direction tuning curves (left axis, green) and tilt tuning curves marginalized over slant and distance (right axis, black) for the four example visual area V3A (V3A) neurons from *Figures 3 and 6*. Data points are mean firing rates and error bars are SEM across trials. Solid green curves are von Mises fits for neurons with significant saccade direction tuning (ANOVA, p<0.05). Black lines are linear interpolations. (**E**) Differences between choice and saccade direction preferences in V3A (N = 124). (**F**) Differences between principal surface tilts and saccade direction preferences for neurons with (colored bars) and without (gray bars) choice-related activity in V3A (left) and caudal intraparietal (CIP) area (right). Bars at ±180° are identical in (**E, F**). (**G**) Comparison of circular variances for the distributions in (**F**). Horizontal lines and asterisks indicate significant differences (two-sample concentration difference test, p<0.05).

The online version of this article includes the following figure supplement(s) for figure 9:

**Figure supplement 1.** Saccade-related activity was associated with less robust 3D tuning in visual area V3A (V3A), but not caudal intraparietal (CIP) area.

Intriguingly, these results indicate that V3A activity predicts the direction and timing of upcoming saccadic eye movements.

We further noted that the V3A time courses were visually more similar to step functions than the CIP time courses, which more closely resembled ramping activity. This informal observation was reflected in steeper growth rates for each quartile in V3A (Q1 = 1.8; Q2 = 1.1; Q3 = 0.82; Q4 = 0.44) than CIP (Q1 = 0.73; Q2 = 0.61; Q3 = 0.56; Q4 = 0.39) and led us to speculate that CIP might temporally integrate saccade-related signals from V3A. For each latency quartile, we therefore integrated the V3A time courses from the onset of V3A activity to when the CIP time courses approximately coalesced (15 ms before saccade initiation; 'Materials and methods'; *Chang et al., 2020b*) and compared them to the observed CIP time courses. Notably, the integrated V3A output (*Figure 8F*, dashed curves) aligned well with the CIP activity (solid curves) for each quartile, as might be expected if CIP accumulates evidence from V3A in favor of particular oculomotor responses. Consistent with CIP being closer to the site of saccade initiation than V3A, the CIP time courses coalesced 15 ms before saccade initiation compared to 59 ms in V3A. These results imply that V3A is not simply a 'visual area' and may suggest that there are parallel visual and saccade-related V3A-to-CIP hierarchies.

We next examined the saccade direction tuning of individual V3A neurons. Saccade direction tuning curves for the four example V3A neurons are shown in *Figure 9A–D* (green curves) along with von Mises fits for those with significant tuning (ANOVA, p<0.05). Across animals, 415 (60%) of the V3A neurons had significant saccade direction tuning (Monkey L: 172, 55%; Monkey F: 154, 58%; Monkey W: 89, 75%), similar to that observed in CIP (across animals: 274, 63%; Monkey L: 153, 70%; Monkey F: 121, 55%) (*Chang et al., 2020b*). Like CIP neurons, the V3A neurons showed parametric tuning for saccade direction with responses that fell off symmetrically from the preferred direction. Correspondingly, the tuning curves were well described by von Mises functions (mean $r$ = 0.92 ± 0.47 × $10^{-3}$ SEM, N = 415).

To test for cross-area differences in saccade direction tuning, we first calculated a saccade discrimination index (SDI; *Equation 2*) for neurons with significant tuning. Surprisingly, saccade direction was more discriminable based on V3A (across animals: mean SDI = 0.49 ± 6.2 × $10^{-3}$ SEM, N = 415; Monkey L: 0.48 ± 0.02, N = 172; Monkey F: 0.51 ± 9.3 × $10^{-3}$, N = 154; Monkey W: 0.48 ± 0.01, N = 89) than CIP (across animals: 0.41 ± 6.2 × $10^{-3}$, N = 274; Monkey L: 0.40 ± 8.7 × $10^{-3}$, N = 153; Monkey F: 0.42 ± 8.7 × $10^{-3}$, N = 121) responses, and the difference was statistically significant (Wilcoxon rank-sum test, across animals: p=4.8 × $10^{-17}$; Monkey L: p=5.1 × $10^{-8}$; Monkey F: p=2.6 × $10^{-11}$). We further compared the tuning bandwidths ($\kappa$ from the von Mises fits) and found that the V3A tuning curves were narrower (across animals: mean $\kappa$ = 6.3 ± 0.26 SEM; Monkey L: 6.1 ± 0.45; Monkey F: 6.6 ± 0.40; Monkey W: 6.2 ± 0.53) than the CIP tuning curves (across animals: 4.7 ± 0.29; Monkey L: 4.4 ± 0.38; Monkey F: 5.0 ± 0.46), and the difference was statistically significant (Wilcoxon rank-sum test, across animals: p=1.9 × $10^{-7}$; Monkey L: p=0.01; Monkey F: p=8.4 × $10^{-5}$). These results are consistent with convergent input from multiple V3A neurons onto individual CIP neurons and parallel the cross-area difference in 3D orientation tuning.

We additionally assessed if choice- and saccade-related activity were functionally dissociable in V3A as they are in CIP (*Chang et al., 2020b*). Across the V3A population, 124 neurons (18%) had both choice- and saccade-related activity. For this subpopulation, choice and saccade preferences generally aligned (*Figure 9E*). The median circular difference (–1.7°) between the preferences was not significantly different from 0° (circular median test, p=0.65). Although the preferences aligned, other characteristics were distinct. First, the saccade tuning curves were narrower (mean $\kappa$ = 6.3 ± 0.26 SEM) than the choice tuning curves (mean $\kappa$ = 4.7 ± 0.46 SEM), and the difference was significant (Wilcoxon rank-sum test, p=6.4 × $10^{-11}$). Second, some neurons carried choice- (48, 7%) or saccade-related (291, 42%) activity only, indicating that the properties were not mutually inclusive. Third, choice-related activity was associated with more robust 3D tuning, whereas saccade-related activity was associated with less robust 3D tuning (*Figure 9—figure supplement 1*), indicating that choice and saccade signals had opposite functional relationships with 3D selectivity in V3A. These results together suggest that the choice- and saccade-related activities were functionally distinct.

## Sensorimotor associations are moderated by choice-related activity

Neurons in CIP form associations between their surface orientation and choice/saccade direction preferences such that the alignment of the tuning curves reflects whether the monkey was trained to report

the near or far side of the plane (*Elmore et al., 2019*; *Chang et al., 2020b*). However, the existence of a similar sensorimotor association was not immediately evident based on the saccade direction and tilt tuning curves (marginalized over slant and distance) of the example V3A neurons (*Figure 9A–D*). Assuming that decision-related processing occupies an intermediate position between sensory and motor activity, it is possible that choice signals have a moderating effect on the sensorimotor association that could obscure the relationship if not taken into consideration. Indeed, we previously found that the strength of the sensorimotor association in CIP depended on the presence of choice signals (*Chang et al., 2020b*).

To test for sensorimotor associations in V3A and evaluate the potential moderating effect of choice signals, we therefore calculated the angular difference between the principal tilt (from the principal orientation) and saccade direction preference for neurons with and without choice-related activity (*Figure 9F*, left, orange and gray bars, respectively). For neurons without choice-related activity, there was no discernible association between the tilt and saccade direction preferences since the distribution was not significantly different from uniform (Rayleigh test, p=0.08, N = 270). However, for neurons with choice-related activity, the distribution was significantly different from uniform (p=$1.2 \times 10^{-18}$, N = 122) and the median circular difference (–1.6°) was not significantly different from 0° (circular median test, p=0.42), revealing a sensorimotor association. Correspondingly, the distribution of preference differences was significantly narrower for neurons with (CV = 0.44) than without (CV = 0.9) choice-related activity (two-sample concentration difference test, p=$1.4 \times 10^{-5}$; *Figure 9G*). Thus, a sensorimotor association was only evident for V3A neurons with choice-related activity.

When we repeated this analysis for CIP, we found a striking difference from V3A (*Figure 9F*, right). Specifically, sensorimotor associations were evident regardless of whether the neurons carried choice signals. The distributions of preference differences were significantly different from uniform both for neurons with (blue bars; p=$3.7 \times 10^{-29}$, N = 134) and without (gray bars; p=$9.7 \times 10^{-5}$, N = 122) choice-related activity. Likewise, the median differences were not significantly different from 0° both for neurons with (–4.2°, p=0.14) and without (–3.2°, p=0.65) choice-related activity. This reveals that choice signals were not necessary for sensorimotor associations in CIP, but the distribution was significantly narrower for neurons with (CV = 0.34) than without (CV = 0.73) choice-related activity (two-sample concentration difference test, p=$2.6 \times 10^{-5}$; *Figure 9G*), indicating that sensorimotor associations were strongest for neurons with choice-related activity.

We further assessed the cross-area difference in the overall strength of sensorimotor associations by comparing the widths of the V3A and CIP preference difference distributions (including neurons with and without choice-related activity). The distribution was significantly broader in V3A (CV = 0.77) than CIP (CV = 0.53; p=$1.2 \times 10^{-3}$), indicating that sensorimotor associations were strongest in CIP (*Figure 9G*). These results thus imply a hierarchical transformation in the strength of sensorimotor associations along the 'where' pathway and suggest a novel role for choice-related activity in sensorimotor processing as explored next.

The above analyses showed that neurons with choice-related activity exhibited more robust 3D pose tuning (*Figure 7*) and formed stronger sensorimotor associations (*Figure 9G*). If choice signals occupy an intermediate position between sensory and motor processing, the strength of sensorimotor associations might depend on the robustness of 3D tuning for neurons with choice-related activity but not those without choice-related activity. In other words, choice-related activity may statistically moderate (*Judd et al., 2014*) the relationship between the robustness of 3D pose tuning and the strength of sensorimotor associations. To test this, we ran a linear regression model where the absolute angular difference between the principal tilt and saccade direction preference depended on tolerance, choice-related activity, and their interaction. Both areas showed a significant interaction (both p≤$1.6 \times 10^{-3}$) such that tolerance had a negligible impact on the strength of the sensorimotor association for neurons without choice-related activity (V3A: slope = 0.16; CIP: slope = –0.15) but a strong impact for neurons with choice-related activity (V3A: slope = –2.2; CIP: slope = –2.5). Thus, this analysis revealed an intricate relationship between sensory, choice, and saccade-related activity in which choice signals moderated the relationship between the robustness of 3D tuning and sensorimotor associations.

## Discussion

Transforming ambiguous 2D retinal images into relevant 3D object representations that can guide action is a fundamental function of the dorsal 'where' pathway. Here, we explicated the parallel processing, hierarchical transformations, and functional associations of visual, choice, and saccade-related signals at the juncture of visual and parietal cortex. Our findings challenge classical notions of sensorimotor dichotomies by revealing that V3A possesses saccade-related activity and defining properties of association cortex, and further implicate choice-related activity as a novel factor in moderating sensorimotor processing.

### Parallel processing, hierarchical transformations, and sensorimotor associations

Multiple lines of evidence converged to support a V3A-to-CIP hierarchy. Focusing first on the visual properties, we found that the median visual response latency in V3A was 6 ms shorter than in CIP and that the receptive fields were smaller in V3A than CIP. At a functional level, 3D orientation and position information (which are confounded in retinal images) were more separable in CIP than V3A, implying a hierarchical resolution of sensory ambiguities that limit the ability to make 3D perceptual inferences.

Intriguingly, our findings may also reveal a V3A-to-CIP hierarchy related to oculomotor processing. While saccade-related activity that predicted the direction and timing of eye movements was prevalent in both areas, it began 6 ms earlier in V3A than CIP (in agreement with the difference in visual response latencies). The finding that the time course of saccade-related activity in CIP closely resembled the temporally integrated V3A output may further suggest that some presaccadic signals originate in a region of visual cortex whose feedforward input includes V1 and V2 (*Zeki, 1980*; *Felleman and Van Essen, 1991*), thus challenging classical notions of sensorimotor dichotomies. This finding is also consistent with CIP accumulating evidence provided by V3A in favor of particular oculomotor responses, which raises the possibility that some of the integration-like properties of LIP (*Shadlen and Newsome, 1996*; *Roitman and Shadlen, 2002*) may reflect bottom-up input (*Lewis and Van Essen, 2000*; *Premereur et al., 2015*; *Van Dromme et al., 2016*). In that case, inactivating CIP, which is architectonically distinct from LIP (*Katsuyama et al., 2010*; *Niu et al., 2020*), might impair saccadic responses during decision-making tasks. Consistent with this, reversible inactivation of CIP during a depth structure categorization task was found to delay saccadic responses (*Van Dromme et al., 2016*). While that delay may have reflected degraded visual discrimination, the current findings also point to the possibility of impaired saccade preparation.

While these experiments revealed saccade-related activity in V3A and CIP, the protocol was not designed to dissociate the potential contributions of visual and presaccadic signals. Nevertheless, several lines of evidence suggest that the observed saccade-related activity cannot be explained by visual responses alone. First, the time course of saccade-related activity was distinct from that of the visual flash response. Examination of the time courses provided some evidence that a presaccadic signal may begin in V3A 7 ms before CIP, consistent with the possibility of a V3A-to-CIP oculomotor hierarchy. Second, there was a pronounced relationship between the growth rate of the saccade-related activity and saccade latency. Third, there was an apparent integrative relationship between the V3A and CIP saccade-related activity, which does not exist for the visual responses. Fourth, the sensorimotor association between tilt and saccade direction preferences as well as the moderation by choice-related activity would not be expected for visual flash responses. The current findings therefore suggest that V3A and CIP may contain presaccadic signals, but a thorough evaluation of this possibility will require future studies.

Many V3A neurons showed functional associations between their orientation, choice, and saccade direction preferences, as in CIP (*Elmore et al., 2019*; *Chang et al., 2020b*). Finding these associations in V3A implies that sensorimotor processing traditionally linked to parietal cortex already occurs in an 'intermediate visual area,' suggesting that V3A may be more appropriately classified as an early association region. Intriguingly, saccade direction discrimination was greater in V3A (where saccade-related activity was associated with poorer 3D pose tuning) while sensorimotor associations were stronger in CIP (where no relationship between saccade-related activity and the robustness of 3D tuning was evident). These cross-area differences may reflect a transition from lower-level representations of saccade-related signals to higher-level sensorimotor associations and parallel the observed changes in visual feature selectivity.

## Visual processing in V3A

Previous studies arrived at categorically different conclusions regarding visual processing in V3A (*Gaska et al., 1987*, *Gaska et al., 1988*; *Galletti and Battaglini, 1989*; *Galletti et al., 1990*; *Sauvan and Peterhans, 1999*; *Nakamura and Colby, 2000*; *Nakamura and Colby, 2002*; *Tsao et al., 2003*; *Anzai et al., 2011*; *Elmore et al., 2019*). Our large V3A sample revealed a heterogeneous population in which neurons ranged from representing low-level visual features to high-level object properties. Earlier discrepancies may reflect a combination of this heterogeneity, small sample sizes, and the possibility of functionally distinct modules within V3A. Another factor that may have contributed to the discrepancies is variability in how V3A was defined (*Nakhla et al., 2021*), highlighting the continued importance of using functional and anatomical localization methods in future investigations of this relatively understudied area.

For some V3A neurons, we found that orientation tuning curve shape (but not gain) was highly tolerant to distance, implying 3D pose tuning. Because the perspective cues in our stimuli were independent of distance, the gain changes must have been driven by stereoscopic cues. This implies that these neurons were selective for gradients of relative disparity. Other V3A neurons showed orientation tuning at a single distance. These neurons may have been selective for absolute disparity gradients, similar to some middle temporal (MT) area neurons (*Nguyenkim and DeAngelis, 2003*). Such neurons may reflect an intermediate stage of visual processing whose outputs are combined to create 3D pose representations. Selectivity for absolute disparity gradients may further account for suppressive effects previously reported within the classical receptive fields of V3A neurons (*Gaska et al., 1987*) since the stimuli were presented at screen distance only and therefore would stimulate portions of the receptive field with non-preferred disparities. Lastly, the orientation tuning curve shape of some V3A neurons was highly distance-dependent, which may reflect tuning for a single absolute disparity. Future studies that perform detailed receptive field submapping of disparity selectivity will be important to explicate the heterogeneity of V3A as well as the transformations by which the visual system achieves 3D pose tuning. One possibility is that absolute disparity representations (in V1) are used to construct absolute disparity gradient detectors (in V3A, MT) and then pose selectivity (in V3A, PIP, CIP).

## Origins and functional implications of choice signals

Contemporary interpretations of choice signals include a mix of feedforward contributions to decision processes, the structure of correlated variability, attention, cognitive and behavioral factors, as well as feedback (*Celebrini and Newsome, 1994*; *Britten et al., 1996*; *Dodd et al., 2001*; *Haefner et al., 2013*; *Gu et al., 2014*; *Ruff and Cohen, 2014*; *Smolyanskaya et al., 2015*; *Cumming and Nienborg, 2016*). Despite these complexities, the current findings are consistent with a feedforward cascade and build-up of choice signals, which may reflect greater contributions of CIP than V3A to 3D perceptual decisions. First, choice-related activity appeared 11 ms earlier in V3A than CIP. Second, it was about twice as prevalent in CIP as V3A. Notably, the structure of correlated variability could not account for the prevalence of choice-related activity either within or across areas. Third, choice-related activity was preferentially carried by neurons with 3D orientation tuning that was more tolerant to distance. This is consistent with reports that neurons that carry choice signals tend to have resolved sensory ambiguities about the information being discriminated, allowing them to more directly contribute to decisions (*Liu et al., 2013*; *Chang et al., 2020b*). Although choice signals may be confounded with feature-based attentional modulation (*Cohen and Newsome, 2008*), this is unlikely to explain our findings. Attention is largely associated with changes in response magnitude (*Reynolds and Heeger, 2009*). However, we found more intricate relationships in which choice-related activity (i) was associated with changes in tuning curve shape that made 3D orientation tuning more tolerant to distance (*Chang et al., 2020b*) and (ii) statistically moderated the strength of sensorimotor associations such that visual and saccade direction preferences were best aligned for neurons that carried choice signals. Indeed, the strength of sensorimotor associations strongly depended on the robustness of 3D pose tuning for neurons with (but not without) choice-related activity. These findings thus reveal a multifaceted landscape of functional associations between sensory, choice-, and motor-related activity. They further imply a novel role for choice signals in sensorimotor processing that might reflect the temporal cascade of sensory processing, response selection, and motor action.

## Materials and methods

### Animal preparation

All procedures followed the National Institutes of Health's Guide for the Care and Use of Laboratory Animals and were approved by the Institutional Animal Care and Use Committee at the University of Wisconsin–Madison (Protocol G005229). Three male rhesus monkeys (*Macaca mulatta*; Monkey L: 6 years of age; Monkey F: 5 years; Monkey W: 5 years) were surgically implanted with a Delrin ring for head restraint and a removable recording grid for guiding electrodes. After recovery, they were trained to sit in a primate chair with head restraint and to fixate visual targets within 2° version and 1° vergence windows for liquid rewards.

### Experimental control and stimulus presentation

Experimental control was performed using the REC-GUI software (RRID:SCR_019008) (*Kim et al., 2019*). Stimuli were rendered using Psychtoolbox 3 (MATLAB R2016b; NVIDIA GeForce GTX 970) and rear-projected onto a polarization preserving screen (Stewart Film Screen, Inc) using a DLP LED projector (PROPixx; VPixx Technologies, Inc) with 1280 × 720 pixel resolution (70° × 43° of visual angle) at 240 Hz. A circular polarizer was used to sequence the presentation of stereoscopic 'half-images' to each eye (120 Hz/eye). Polarized glasses were worn. A phototransistor circuit was used to confirm the synchronization of the left and right eye images as well as align neuronal responses to the stimulus onset. Eye tracking was performed optically at 1 kHz (EyeLink 1000 plus, SR Research). The monkeys sat 57 cm from the screen.

### Visual stimuli

The stimuli were previously described in detail (*Chang et al., 2020a*; *Chang et al., 2020b*). Briefly, planar surfaces subtending 20° of visual angle were presented at the center of the screen. They were defined by 250 nonoverlapping dots that were uniformly distributed across the plane and rendered with stereoscopic and perspective cues. Surface orientation was described using tilt and slant (*Stevens, 1983*; *Rosenberg et al., 2013*). All combinations of eight tilts (0–315°, 45° steps) and four slants (15–60°, 15° steps) plus the frontoparallel plane (tilt undefined, slant = 0°) were presented (N = 33). All orientations were presented at four distances (37, 57, 97, and 137 cm; N = 132 unique poses). The dots were scaled with distance such that their screen size only depended on slant. At a slant of 0°, each dot subtended 0.35°.

The fixation point subtended 0.3° and was always at 57 cm (screen distance). Keeping its distance constant while varying the plane's distance was a key design feature (*Nguyenkim and DeAngelis, 2003*; *Hegdé and Van Essen, 2005*; *Ban and Welchman, 2015*; *Alizadeh et al., 2018*; *Henderson et al., 2019*) that conferred two advantages over yoking the distance of the stimulus and fixation point (*Banks et al., 2001*; *Hillis et al., 2004*). First, it ensured that the monkeys could not rely on local absolute disparity cues to judge 3D orientation (*Elmore et al., 2019*; *Chang et al., 2020a*; *Chang et al., 2020b*). Second, it allowed us to dissociate the effects of stimulus distance and vergence signals (which would have otherwise been confounded) on the neuronal responses, which is important because V3A and CIP carry extraretinal signals.

### Experimental protocol

#### Tilt discrimination task

The monkeys performed an 8AFC tilt discrimination task (*Chang et al., 2020a*; *Chang et al., 2020b*). Each trial began by fixating a circular target at the center of the screen for 300 ms. A planar surface then appeared for 1 s. The fixation target and plane then disappeared and eight choice targets corresponding to the eight possible tilts appeared at polar angles of 0–315° in 45° steps (11° eccentricity). The nearest side of the plane was indicated by making a saccade to the corresponding target (e.g., the right target for a right-near plane) in exchange for a liquid reward for correct responses. Because frontoparallel planes were task ambiguous (tilt is undefined at slant = 0°), responses to those stimuli were rewarded with equal probability (12.5%).

#### Visually guided saccade task

The monkeys also performed a visually guided (pop-up) saccade task (*Chang et al., 2020b*). The timing was matched to the tilt discrimination task. Each trial began by fixating a target (identical to

the target in the tilt discrimination task) at the center of the screen for 1.3 s. The fixation target then disappeared and a single saccade target appeared at one of the eight choice target locations. A saccade to the target was made for a liquid reward.

The two tasks were interleaved within a block design. For all CIP and 422 V3A neurons, each block included one completed trial for each of the following: (i) tilt discrimination task: (8 tilts × 4 non-zero slants + 8 frontoparallel planes) × 4 distances (160 trials) and (ii) saccade task: 8 directions × 4 repeats (32 trials). For 270 V3A neurons, each saccade direction was repeated once per block. Trials were aborted and the data discarded if fixation was broken before the choice/saccade target(s) appeared, or if a response was not provided within 500 ms. A minimum of five blocks was required to include a neuron for analysis.

## Behavioral data analysis

Tilt discrimination performance was quantified by calculating the distribution of reported tilt errors (ΔTilt = reported tilt – presented tilt) and fitting a von Mises probability density function:

$$VM(\Delta\text{Tilt}) = \frac{e^{\kappa \cdot cos(\Delta\text{Tilt} - \mu)}}{2\pi \cdot I_0(\kappa)}. \tag{1}$$

The mean ($\mu$) and concentration ($\kappa$) parameters describe the accuracy and sensitivity, respectively (*Seilheimer et al., 2014*; *Dakin and Rosenberg, 2018*). Values of $\mu$ closer to 0° indicate greater accuracy and larger $\kappa$ indicate greater sensitivity. An upper bound of $\kappa = 18$ was set in the estimation routine (*Chang et al., 2020a*; *Chang et al., 2020b*). A modified Bessel function of order 0, $I_0(\kappa)$, normalizes the function to have unit area.

## Neuronal recordings

To target the areas, magnetic resonance imaging (MRI) scans were collected on a 3-Tesla GE MR750 scanner before and after implanting the head restraint ring. To estimate the penetration trajectories, the CARET software was used to register the structural scans to the F99 atlas (*Van Essen et al., 2001*) and align the recording grid (*Rosenberg et al., 2013*; *Rosenberg and Angelaki, 2014a*; *Rosenberg and Angelaki, 2014b*; *Chang et al., 2020b*). During penetrations, observed gray/white matter and sulcal transitions were referenced to the MRI. Area CIP is located in the caudal portion of the lateral bank of the intraparietal sulcus and ventral to LIP. Area V3A is located adjacent to and ventral-laterally to CIP. There is a swath of white matter dorsal to V3A and lateral to CIP.

The order of the recording sessions went as follows. First, 41 V3A sessions were performed (Monkey L: N = 21; Monkey F: N = 20). Then, all 53 CIP sessions were performed (Monkey L: N = 26; Monkey F: N = 27). Finally, 50 more V3A sessions were performed (Monkey L: N = 18; Monkey F: N = 18; Monkey W: N = 14). The recordings were performed using linear array probes with either four or eight tetrodes (NeuroNexus, Inc). The tetrodes were separated by 300 µm and electrodes within a tetrode were separated by 25 µm. Neuronal signals (sampled at 30 kHz) along with eye position and phototransistor signals (each sampled at 1 kHz) were stored using a Scout Processor (Ripple, Inc). Tetrode-based spike sorting was performed offline using the KlustaKwik (K. Harris) semi-automatic clustering algorithm in MClust (MClust-4.0, A.D. Redish et al.) followed by manual refinement using Offline Sorter (Plexon, Inc). Only well-isolated single neurons verified by at least two authors were included.

## Receptive field mapping and analysis

Initial receptive field (RF) estimates were made by hand-mapping with patches of random dots, sinusoidal gratings, and/or orientated bars. During hand-mapping, fixation was maintained on a target (0.3°) at the center of the screen and a liquid reward was provided every 2–3 s of continuous fixation. Breaks in fixation resulted in the disappearance of the fixation point and stimulus for 1 s. An automated stimulus in which bright and dark squares were flashed one square at a time was then presented on a gray background. For V3A, 2° × 2° squares tiled a region centered on and larger than the hand estimate. For CIP, the squares were 4° × 4° and tiled the entire screen. A monkey was required to first fixate a target (0.3°) at the center of the screen for 300 ms. Then, while maintaining fixation, the squares were flashed (150 ms duration) in an alternating sequence at pseudorandom locations. A liquid reward was provided after every 20 flashes. If fixation was broken, the monkey was required to again fixate the target for 300 ms before the sequence resumed. A single repetition was

completed when both bright and dark squares had been flashed at every location. At least five repetitions were collected each session.

The RF boundary was estimated offline by calculating the firing rate at each tiled location, averaging over the bright and dark square responses. The responses at each location were then compared to baseline, calculated using the last 150 ms of the fixation periods preceding the stimulus sequences, to identify significant responses (ANOVA, p<0.05). The RF map was then manually smoothed and an envelope contour drawn. This procedure produced RF estimates for 355 (51%) V3A and 112 (26%) CIP neurons. The cross-area difference in the proportion of neurons for which a RF could be estimated may reflect that the mapping stimulus was likely more appropriate for lower- than higher-level visual areas.

## Neuronal data analyses

### Visual response latency

For each neuron, spike trains were aligned to the stimulus onset using the phototransistor signal. Each spike train (1 ms bins) was convolved with a double exponential function and then averaged across trials to create spike density functions (SDFs) (*Chang et al., 2020b*). A neuron's visual latency was defined as the first time point after the stimulus onset where the SDF significantly deviated (ANOVA followed by Tukey's HSD test, p<0.05) from the baseline activity (calculated using the last 150 ms of the fixation periods preceding the stimuli) for at least 30 ms. Firing rates were calculated from the area's median visual latency to the end of the stimulus presentation.

### Discrimination indices (DIs)

We calculated DIs to quantify how well preferred and non-preferred conditions could be discriminated from single-neuron responses (*Prince et al., 2002*):

$$\text{DI} = \frac{R_{max} - R_{min}}{R_{max} - R_{min} + 2\sqrt{\frac{SSE}{N-M}}} \tag{2}$$

where $R_{max}$ and $R_{min}$ are the maximum and minimum mean responses across the tuning curve, $SSE$ is the sum squared error around the mean responses for each condition, $N$ is the total number of trials, and $M$ is the number of conditions. For neurons with large response modulation and low response variability, responses to preferred and non-preferred stimuli will be highly discriminable, and the DI correspondingly closer to one. Otherwise, the DI will be closer to zero. Four DIs were computed following *Equation 2*. The SODI was used to quantify how well the 3D orientation (over all slant–tilts) could be discriminated at each distance (M = 33). The TDI was used to quantify how well the tilt could be discriminated at given slant–distance combinations (M = 8). The CDI was used to quantify how well the choice could be discriminated (M = 8). SDI was used to quantify how well the saccade direction could be discriminated (M = 8).

### Quantifying the dependency of orientation tuning on distance

The tolerance of the 3D orientation tuning curves to distance was quantified as previously described (*Chang et al., 2020b*). Briefly, we fit the 3D pose (orientation × distance) tuning curve with a multiplicatively separable model:

$$R(\theta, D) = DC + g \cdot H(\theta) \cdot F(D) \tag{3}$$

where $R$ is the response to orientation θ (tilt and slant) and distance $D$, $DC$ is an offset, $g$ sets the response amplitude, $H(\theta)$ is the orientation tuning curve, and $F(D)$ is the distance tuning curve. A tolerance index quantifying the dependence of the 3D orientation tuning curve shape on distance was calculated as the average correlation between the observed tuning curve and fit at each distance. We additionally tested an additively separable model $R(\theta, D) = DC + H(\theta) + F(D)$, but did not present the results because the multiplicative model better described the responses of 690/692 V3A and 437/437 CIP neurons.

We also quantified the extent to which orientation preferences differed across distance. For each neuron, the preferred orientation was estimated at each distance with significant orientation selectivity (ANOVA, p<0.05; Bonferroni–Holm corrected for four distances) by fitting a Bingham function

(*Rosenberg et al., 2013*). The principal orientation, the axis about which the preferences clustered, was determined by arranging the surface normal vectors corresponding to the preferences into a matrix and calculating the eigenvectors. The principal orientation was defined by the eigenvector with the largest eigenvalue (*Chang et al., 2020b*).

## Choice-related activity

Choice-related activity was calculated using frontoparallel plane trials only. In every session, each monkey chose every choice target, and the choice distributions were broad (Monkey L: mean circular variance = 0.79 ± 0.09 standard deviation, N = 65 sessions; Monkey F: 0.82 ± 0.09, N = 65; Monkey W: 0.64 ± 0.14, N = 14). Importantly, there were no significant correlations between the mean choices of the monkeys and the choice preferences of the simultaneously recorded neurons (Monkey L: $r$ = 0.13, p=0.24; Monkey F: $r$ = –0.11, p=0.08; Monkey W: $r$ = 0.21, p=0.18).

Choice tuning was assessed from the onset of choice-related activity for the area (calculated in an iterative fashion following *Chang et al., 2020b*) to the end of the stimulus presentation, as briefly described here. Response differences associated with the plane's distance were removed by z-scoring the baseline-subtracted responses separately for each distance. For neurons with significant choice tuning (ANOVA, p<0.05), responses were then grouped according to the tilt choice. For each tuned neuron, the average SDF was then computed for each choice direction and labeled relative to the neuron's preferred choice (preferred choice, ±45°, ±90°, ±135°, and 180°). The SDFs were then averaged across neurons to create eight population-level time courses. The onset was defined as the first time point that the time courses significantly differed (ANOVA, p<0.05) for at least 30 ms (*Rosenberg et al., 2013*). This process was iteratively performed until the onset no longer changed.

## Saccade-related activity

Saccade onset was defined as the first time point that the velocity of either eye was ≥150°/s. Saccade direction tuning was assessed from the start of saccade-related activity for the area to the saccade onset. A neuron was classified as having saccade-related activity if the baseline-subtracted firing rates significantly depended on the saccade direction (ANOVA, p<0.05). The onset of saccade-related activity was calculated following the procedure described above for the choice-related activity (*Chang et al., 2020b*).

The time point at which the time courses of saccade-related activity conditioned on saccade latency approximately coalesced was defined as when the sum squared error between the time courses and their mean was smallest (*Chang et al., 2020b*). To visually compare the temporally integrated V3A and CIP time courses, we applied a DC offset and multiplicative gain to each integrated V3A time course to minimize the sum squared error with the CIP time course.

## Noise and signal correlations

We calculated noise and signal correlations between pairs of simultaneously recorded neurons on the same tetrode in V3A (N = 404 pairs) and CIP (N = 244 pairs). To calculate noise correlations, we took the spike counts across trials for each 3D pose condition (N = 132) and removed outliers (>3 standard deviations from the mean response) (*Zohary et al., 1994*; *Kohn and Smith, 2005*; *Huang and Lisberger, 2009*; *Gu et al., 2011*). To remove stimulus-dependent response differences, the remaining spike counts were z-scored separately for each condition. The correlation between the two neurons' responses was then computed across conditions and trials. Signal correlations were computed between the 3D pose tuning curves.

## Vergence control

To determine if the 3D pose responses were significantly affected by small vergence eye movements that did not violate the vergence window, we performed an ANCOVA to test for main effects of stimulus tuning with vergence included as a covariate. Tilt (linearized into cosine and sine components), slant, and distance were independent factors and the mean vergence was a covariate. Only 67 (9.7%) V3A neurons showed a significant effect of vergence (p<0.05). Importantly, the significance of the main effects did not depend on whether vergence was included as a covariate for all but 13 (1.9%) V3A neurons. These results are comparable to those in the CIP data (*Chang et al., 2020b*) and other

studies (*DeAngelis and Uka, 2003*; *Elmore et al., 2019*), and suggest that vergence errors had a minimal impact on the findings.

## Acknowledgements

We thank Zikang Zhu for helpful discussion, as well as Meghan Lowe and Satchal Postlewaite for help with spike sorting. Further support was provided by the National Institutes of Health Grant P51OD011106 to the Wisconsin National Primate Research Center and the National Institute of Child Health and Human Development Grant P50HD105353 to the Waisman Center.

## Additional information

### Funding

| Funder | Grant reference number | Author |
|---|---|---|
| National Institutes of Health | T32EY027721 | Raymond Doudlah |
| National Science Foundation | DGE-1545481 | Raymond Doudlah Lowell W Thompson |
| National Institutes of Health | T32NS105602 | Lowell W Thompson |
| McPherson Eye Research Institute | Graduate Student Support Initiative | Lowell W Thompson |
| Alfred P. Sloan Foundation | FG-2016-6468 | Ari Rosenberg |
| Whitehall Foundation | 2016-08-18 | Ari Rosenberg |
| Greater Milwaukee Foundation | Shaw Scientist Award | Ari Rosenberg |
| National Institutes of Health | EY029438 | Ari Rosenberg |

The funders had no role in study design, data collection and interpretation, or the decision to submit the work for publication.

### Author contributions

Raymond Doudlah, Conceptualization, Data curation, Software, Formal analysis, Funding acquisition, Validation, Investigation, Visualization, Methodology, Writing - original draft, Writing – review and editing; Ting-Yu Chang, Conceptualization, Data curation, Software, Formal analysis, Funding acquisition, Validation, Investigation, Visualization, Methodology, Writing – review and editing; Lowell W Thompson, Software, Formal analysis, Funding acquisition, Validation, Writing – review and editing; Byounghoon Kim, Conceptualization, Resources, Data curation, Software, Formal analysis, Supervision, Validation, Methodology, Project administration, Writing – review and editing; Adhira Sunkara, Conceptualization, Resources, Software, Methodology, Writing – review and editing; Ari Rosenberg, Conceptualization, Resources, Data curation, Software, Formal analysis, Supervision, Funding acquisition, Validation, Investigation, Visualization, Methodology, Writing - original draft, Project administration, Writing – review and editing

### Author ORCIDs

Raymond Doudlah (iD) http://orcid.org/0000-0003-3631-5947
Ting-Yu Chang (iD) http://orcid.org/0000-0003-3964-0905
Byounghoon Kim (iD) http://orcid.org/0000-0001-7159-5134
Ari Rosenberg (iD) http://orcid.org/0000-0002-8606-2987

### Ethics

This study was performed in strict accordance with the recommendations of the National Institutes of Health's Guide for the Care and Use of Laboratory Animals. All experimental procedures and

surgeries were approved by the Institutional Animal Care and Use Committee (IACUC) at the University of Wisconsin-Madison (Protocol #: G005229).

### Decision letter and Author response
Decision letter https://doi.org/10.7554/eLife.78712.sa1
Author response https://doi.org/10.7554/eLife.78712.sa2

---

## Additional files

### Supplementary files
• Transparent reporting form

### Data availability
All data generated or analyzed during this study are available through the Open Science Framework via our lab's profile at https://osf.io/8wxk7/.

The following dataset was generated:

| Author(s) | Year | Dataset title | Dataset URL | Database and Identifier |
| --- | --- | --- | --- | --- |
| Rosenberg A, Doudlah R, Chang T-Y, Thompson L, Kim B | 2022 | Parallel processing, hierarchical transformations, and sensorimotor associations along the 'where' pathway | https://osf.io/a89gx/ | Open Science Framework, a89gx |

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

## Appendix 1

Here, we show that the discrimination index is invariant to linear transformations of the neuronal responses using the example of z-scoring. The discrimination index is defined as

$$DI = \frac{R_{max} - R_{min}}{R_{max} - R_{min} + 2\sqrt{\frac{SSE}{N-M}}}.$$

where $R_{max}$ and $R_{min}$ are the maximum and minimum mean responses across the tuning curve, $SSE$ is the sum squared error around the mean responses for each condition, $N$ is the total number of trials, and $M$ is the number of conditions.

The z-score transformation is defined as

$$Z(x) = \frac{x - \mu}{\sigma}.$$

where $x$ is the observed value, $\mu$ is the sample mean, and $\sigma$ is the sample standard deviation. We define $DI_Z$ as the discrimination index of the z-scored responses:

$$DI_Z = \frac{\frac{R_{max} - \mu}{\sigma} - \frac{R_{min} - \mu}{\sigma}}{\frac{R_{max} - \mu}{\sigma} - \frac{R_{min} - \mu}{\sigma} + Z\left(2\sqrt{\frac{SSE}{N-M}}\right)} = \frac{R_{max} - R_{min}}{R_{max} - R_{min} + \sigma \cdot Z\left(2\sqrt{\frac{SSE}{N-M}}\right)}$$

Thus, if $Z\left(2\sqrt{\frac{SSE}{N-M}}\right) = \frac{1}{\sigma} \cdot 2\sqrt{\frac{SSE}{N-M}}$, then $DI = DI_Z$. The $SSE$ is given by

$$SSE = \sum_{i=1}^{M} \sum_{j=1}^{n_i} \left(R_{ij} - \mu_i\right)^2$$

where $M$ is the total number of conditions, $n_i$ is the number of trials for the $i$th condition, $R_{ij}$ is the response for the $i$th condition and $j$th trial, and $\mu_i$ is the mean response for the $i$th condition. Since $\sum_{i=1}^{M} n_i = N$,

$$Z\left(2\sqrt{\frac{SSE}{N-M}}\right) = Z\left(2\sqrt{\frac{\sum_{i=1}^{M} \sum_{j=1}^{n_i} \left(R_{ij} - \mu_i\right)^2}{N-M}}\right)$$

$$= 2\sqrt{\frac{\sum_{i=1}^{M} \sum_{j=1}^{n_i} \frac{1}{\sigma^2}\left(R_{ij} - \mu_i\right)^2}{N-M}} = \frac{1}{\sigma} \cdot 2\sqrt{\frac{\sum_{i=1}^{M} \sum_{j=1}^{n_i} \left(R_{ij} - \mu_i\right)^2}{N-M}}$$

Thus, $Z\left(2\sqrt{\frac{SSE}{N-M}}\right)$ reduces to $\frac{1}{\sigma} \cdot 2\sqrt{\frac{SSE}{N-M}}$, and $DI = DI_Z$.

