## [Editor Report]

This study compares the neuronal activity of two interconnected cortical areas in the dorsal visual pathway, V3A and CIP, during perceptual decisions based on judging the tilt of 3D visual patterns. This is a novel comparison between neural activity in these two cortical areas during perceptual decisions and gives insight into the hierarchical relationship and roles of these two areas. CIP shows higher-order spatial representations and more choice-correlated responses. Furthermore, the study finds modulation of V3A activity by extraretinal factors, suggesting that V3A be better characterized as contributing to higher-order behavioral functions as opposed to low-level visual feature processing.

---

## [Decision Letter]

**Decision letter after peer review:**

Thank you for submitting your article "Parallel processing, hierarchical transformations, and sensorimotor associations along the 'where' pathway" for consideration by *eLife*. Your article has been reviewed by 3 peer reviewers, one of whom is a member of our Board of Reviewing Editors, and the evaluation has been overseen by Joshua Gold as the Senior Editor. The following individual involved in the review of your submission has agreed to reveal their identity: Peter Janssen (Reviewer #3).

Essential revisions:

(1) R1: The manuscript should provide more information and analysis of behavioral performance in V3A and CIP sessions, to address the possibility that any differences in neuronal results observed between areas can be explained based on the performance of the animals.

(2) R1: The methods should be expanded to include additional information about the sequence and timing of V3A and CIP experimental sessions. For example, were all CIP sessions conducted first followed by V3A sessions, or were they interleaved?

(3) R1: The overall presentation of the results would be improved by adding information about the animal to animal variability in the results, and clarifying whether examples are shown from different animals (e.g. Figures 1, 2, 3)

4) R1: The presentation of results and rationale for pooling data between animals can be improved as described by point #4 from reviewer 1.

(5) R1: Regarding the analysis of presaccadic activity in V3A, it would be good to show that the results are robust to the assumptions and parameters of the analysis and justify the use of ANOVA as the statistical test.

6. R2 questions using the visually guided saccade to dissociate visual and saccade-related responses. The approach should be clarified regarding how "CIP time courses approximately coalesced".

7. R2: The discrimination index should be described more clearly, including understanding what kind of index values are expected under different conditions.

8. R2 suggests clarifying the section entitled, "Hierarchical refinement of 3D pose representation."

9. R2 suggests refining and clarifying the analysis of choice-related activity by assessing the amount of behavioral bias in the monkeys' choices.

(10) It is suggested by R3 to temper claims that the results are strong evidence for a hierarchical relationship between the two brain areas, and to place more emphasis on the novel results of pre-saccadic activity in V3A.

11) R3 recommends dropping statements regarding the conflict between anatomical and functional data, as it is not judged to be critical for the current study.

12) R3: Clarify whether the difference in timing of the onset of choice activity is significantly different between CIP and V3A.

*Reviewer #1 (Recommendations for the authors):*

The study would benefit from additional details about the experimental methods to understand the temporal relationship between recordings from V3A and CIP. Revisions to the analysis approach, as well as breaking down the analysis and results on a per animal basis, would strengthen the study as well.

(1) The CIP and V3a data were obtained from different recording sessions. The manuscript should provide more information and analysis of behavioral performance in V3A and CIP sessions, to address the possibility that any differences in neuronal results observed between areas can be explained based on the performance of the animals.

(2) Related to the comment above, additional information about the sequence and timing of V3A and CIP sessions should be included in the manuscript. For example, were all CIP sessions conducted first followed by V3A sessions, or were they interleaved?

(3) A general comment regarding the presentation of the results is that there is a notable lack of information about the animal to animal variability in the results, and in many cases, data is shown for only one animal or is not specified regarding whether data from multiple animals are shown.

(3a) Figures 1 & 2 show imaging and behavioral data for only one of the monkeys. In both cases, data/results should be shown for all animals.

(3b) In figure 3, it should be made clear which animals each of the single neuron examples are from.

4) Related to point 3 – for many (but not all) of the population level analyses, data is apparently combined across animals and stats reported on the pooled data. This is not unusual and generally fine, but appropriate tests need to be done and reported regarding whether results were consistent between animals, and how they differ. It is not clear what rationale is being followed to determine which analyses are reported separately for each animal, and which are from pooled data. Also, it should be reported whether findings were statistically significant within each animal, or only when data is combined across animals. The study has recorded sizeable populations of neurons within each animal and brain area, so the data seems likely to support a more granular level of analysis animal by animal.

(5) Regarding the analysis of presaccadic activity in V3A, it would be good to show that the results are robust to the assumptions and parameters of the analysis. For example, picking the divergence time based on an ANOVA at P<0.05 seems like a liberal criterion that could be prone to noise. Also, given the parametric tuning, is ANOVA really the most appropriate choice for a test?

*Reviewer #2 (Recommendations for the authors):*

I have a few suggestions to make, which I think, would improve the article and its clarity.

1. As stated in the public review, I don't think visually guided saccades allow you to dissociate from visual and saccade-related responses. Moreover, I don't understand how you compute how "CIP time courses approximately coalesced".

2. I think the rationale between DI and how they behave could be better explained. I am not sure I truly understand how it is supposed to behave with different levels of response relative to different variability.

3. I'm a bit confused by the paragraph named Hierarchical refinement of 3D pose representation. Here you show that 3D orientation tuning is less distance dependent in CIP than in V3a (Figure 4a). I don't understand what is the difference with the following analysis of angular deviation.

4. In the analysis of choice-related activity, you focus on trials with 0 degrees tilt and use animals' behavior to infer how they interpret these ambiguous stimuli. The analysis you propose assumes that animals' choices are evenly distributed among the 8 possible directions. Are animals biased-free?

*Reviewer #3 (Recommendations for the authors):*

I am not convinced that the correspondence between the integrated V3A time courses and the CIP time courses really reflects a parallel hierarchy. It is also possible that these presaccadic signals originating from a higher area arrive differently in the two areas. In any case, the manuscript would be improved if the authors would de-emphasize and maybe reduce the first part of the results, and highlight more the novel results on pre-saccadic activity in V3A.

The authors try to rescue the fact that many results are expected by suggesting an apparent conflict between anatomical and functional data, but the lower selectivity for 3D orientation in CIP compared to V3A as reported in the Elmore study is a very small effect (SODI of 0.63 vs 0.68) with a small sample of V3A neurons which could be explained by other experimental factors. I would strongly recommend dropping these statements in the introduction and discussion (also on p.9) since it is not really necessary for this study.

It is not clear whether the difference in timing of the onset of choice activity is significantly different between CIP and V3A.

---

## [Author Response]

Essential revisions:1) R1: The manuscript should provide more information and analysis of behavioral performance in V3A and CIP sessions, to address the possibility that any differences in neuronal results observed between areas can be explained based on the performance of the animals.

This is an excellent point that was not addressed in the first submission. We updated the analysis to test for a difference in performance between the V3A and CIP sessions. The analysis confirmed that there was no significant difference in performance between these sessions (lines: 143-149).

(2) R1: The methods should be expanded to include additional information about the sequence and timing of V3A and CIP experimental sessions. For example, were all CIP sessions conducted first followed by V3A sessions, or were they interleaved?

As suggested, we now report the sequence of the V3A and CIP recording sessions (lines: 898-901). Briefly, we first performed 41 V3A sessions (Monkeys L and F), then all 53 CIP sessions (Monkeys L and F), and then 50 additional V3A sessions (Monkeys L, F, and W).

(3) R1: The overall presentation of the results would be improved by adding information about the animal to animal variability in the results, and clarifying whether examples are shown from different animals (e.g. Figures 1, 2, 3)

We completely agree. For all main analyses, we added statistics for each animal to complement the population results. Where relevant, we now discuss between-animal variability (which was minimal). Analyses that distinguished whether neurons carried choice-related activity were performed at the population-level only because some of the individual sample sizes became rather small (see lines: 399-402). We now indicate from which animal each example neuron came (lines: 192-193).

4) R1: The presentation of results and rationale for pooling data between animals can be improved as described by point #4 from reviewer 1.

We agree that this was not clear in the first submission. To clarify, we ran behavioral analyses as well as comparisons between the behavioral and neuronal responses separately for each animal. The neuronal analyses were run at the population level. As indicated in our response to comment #3, the resubmission includes neuronal analyses for the individual animals as well as pooled across animals.

(5) R1: Regarding the analysis of presaccadic activity in V3A, it would be good to show that the results are robust to the assumptions and parameters of the analysis and justify the use of ANOVA as the statistical test.

We thank the reviewer for bringing up this point. In the first submission, we did not clearly indicate that the onset time was defined as the first time point at which the time courses significantly diverge (ANOVA, p < 0.05) for at least 30 consecutive ms. The 30 ms criterion is widely used when calculating latencies because it makes the estimates more conservative and eliminates false positives. We have clarified this in the Materials and methods section (lines: 937-940 and 994-995).

We further verified the robustness of the results in two ways. First, we recalculated the onset of choice- and saccade-related activity using an ANOVA with more conservative parameters (significance values of 0.01, 0.001, and 0.0001). As expected, the onsets shifted to later times, but always by roughly equal amounts in V3A and CIP. In all cases the cross-area latency difference was within 2 ms of the reported difference, and the conclusions regarding the cross-area differences never changed. Second, we recalculated the onset of choice- and saccade-related activity using a Kruskal-Wallis test (nonparametric). In all cases, the onset times were within 3 ms of the times calculated using an ANOVA, and the conclusions regarding the cross-area differences did not change.

We also used an ANOVA to test if the responses of individual neurons depended on the saccade direction, which allowed for a direct comparison with previous work. Using a Kruskal-Wallis test (nonparametric), the classification (tuned vs. not tuned) was the same for 1053/1129 (93%) of the neurons. Additionally, we note that the saccade direction tuning curves of neurons with significant tuning were well described by von Mises functions (mean r = 0.92 ± 0.47x10^-3^ SEM), which would not be expected if they were not accurately classified using the ANOVA.

6. R2 questions using the visually guided saccade to dissociate visual and saccade-related responses. The approach should be clarified regarding how "CIP time courses approximately coalesced".

This is an important point which should have been addressed more thoroughly in the first submission. In the resubmission, we discuss that the current study cannot unambiguously distinguish the contributions of visual and presaccadic activity to the observed saccade-related activity (lines: 733-734). To be more agnostic about the possibility of presaccadic activity and to match the terminology used in Chang et al. (2020b), we now use the term “saccade-related activity”. We also added a supplemental figure which shows that the saccade-related activity follows a distinct time course from the visual flash responses measured during receptive field mapping (Figure 8―figure supplement 1). In the Discussion, we additionally highlight that several of the findings suggest that the observed saccade-related activity cannot be attributed to visual responses, and that future experiments will be required to thoroughly dissociate the contributions of visual and presaccadic activity in V3A/CIP (lines: 735-745).

In the Results section (lines: 565-566), we added additional pointers to the Materials and methods section (lines: 1003-1007) where we describe how we calculated the coalescent points for both the V3A and CIP time courses. At the same locations, we reference Chang et al. (2020b) which also used this method.

7. R2: The discrimination index should be described more clearly, including understanding what kind of index values are expected under different conditions.

As suggested, we expanded the description of the discrimination index (DI) to clarify its interpretation and how the mean and variance of the neuronal responses together determine the value (lines: 256-257 and 949-951).

8. R2 suggests clarifying the section entitled, "Hierarchical refinement of 3D pose representation."

We thank the reviewer for this suggestion. We have clarified that the two analyses in this section are complementary to one another in that they provide an assessment of how the overall shape of the orientation tuning curve depends on distance (Tolerance) and how the preferred orientation (independent of tuning bandwidth) depends on distance (lines: 312-334). Together, they provide convergent evidence of a transformation from lower-level visual feature selectivity in V3A to higher-level 3D object representations in CIP.

9. R2 suggests refining and clarifying the analysis of choice-related activity by assessing the amount of behavioral bias in the monkeys' choices.

This is an excellent point that we did not address in the first submission. We now report that each of the monkeys chose all targets in every session and provide summary statistics which show that the choice distributions were very broad. We further report that the session-by-session mean choices of the monkeys and the preferred choice directions of the neurons were not significantly correlated for any monkey, suggesting that their choices were not associated with the neuronal choice preferences (lines: 980-985).

(10) It is suggested by R3 to temper claims that the results are strong evidence for a hierarchical relationship between the two brain areas, and to place more emphasis on the novel results of pre-saccadic activity in V3A.

This is a great point. As suggested, we tempered our interpretation of the differences in saccade-related activity between V3A and CIP as evidence of a hierarchy and focus more on the novel result of saccade-related activity in V3A.

(11) R3 recommends dropping statements regarding the conflict between anatomical and functional data, as it is not judged to be critical for the current study.

We agree that this point is secondary to the goals of the study and removed all such statements.

(12) R3: Clarify whether the difference in timing of the onset of choice activity is significantly different between CIP and V3A.

To statistically compare the V3A and CIP onsets of choice- and saccade-related activity at the population level, it was necessary to perform permutation tests based on bootstrapped values. These individual statistics were not significant, but the latency differences were highly consistent across all domains, supporting the proposed hierarchy. We now highlight in the Discussion that V3A activity preceded CIP activity by a similar amount in every domain: visual onset (6 ms; lines: 710-712), choice-related activity onset (11 ms; lines: 793-794), saccade-related activity onset aligned to the saccade initiation (6 ms; lines: 717-719), and inflection point in the time course of the saccade-related activity aligned to the target onset (7 ms; lines: 737-739). Notably, these analyses were performed using different neuronal subpopulations and experimental trials, providing convergent evidence of a V3A to CIP hierarchy.